# No time to train! Training-Free Reference-Based Instance Segmentation

## Abstract

The performance of image segmentation models has historically been constrained by the high cost of collecting large-scale annotated data. The Segment Anything Model (SAM) alleviates this original problem through a promptable, semantics-agnostic, segmentation paradigm and yet still requires manual visual-prompts or complex domain-dependent prompt-generation rules to process a new image. Towards reducing this new burden, our work investigates the task of object segmentation when provided with, alternatively, only a small set of reference images. Our key insight is to leverage strong semantic priors, as learned by foundation models, to identify corresponding regions between a reference and a target image. We find that correspondences enable automatic generation of instance-level segmentation masks for downstream tasks and instantiate our ideas via a multi-stage, training-free method incorporating (1) memory bank construction; (2) representation aggregation and (3) semantic-aware feature matching. Our experiments show significant improvements on segmentation metrics, leading to state-of-the-art performance on COCO FSOD (36.8% nAP), PASCAL VOC Few-Shot (71.2% nAP50) and outperforming existing training-free approaches on the Cross-Domain FSOD benchmark (22.4% nAP).

## 1 Introduction

It is well understood that collecting large-scale annotations for segmentation tasks is a costly and time-consuming process Benenson et al. (2019); Papadopoulos et al. (2021); Gupta et al. (2019). Recent advances in promptable segmentation frameworks Zou* et al. (2023); Wang et al. (2023b); Jiang et al. (2024); Wang et al. (2023c); Pan et al. (2023); Liu et al. (2024a), epitomised by the Segment Anything Model (SAM) Kirillov et al. (2023); Ravi et al. (2024), have significantly reduced manual effort by enabling high-quality mask generation using simple geometric prompts such as points, boxes or rough sketches. While this represents a substantial advancement in reducing manual effort, these masks lack semantic awareness Shin et al. (2024); Espinosa et al. (2024); Han et al. (2023); Ji et al. (2024) and require either manual intervention or complex, domain-specific prompt-generation pipelines to function autonomously (e.g., medical imaging Ma et al. (2024a); Lei et al. (2023); Zhang et al. (2024c); Zhang & Jiao (2023), agriculture Carraro et al. (2023); Tripathy et al. (2024), remote sensing Osco et al. (2023); Ma et al. (2024b); Wang et al. (2023a)). Relying on manual prompts for each image limits their scalability (especially for large datasets or scenarios requiring automatic processing), while relying on domain-constrained automated pipelines restricts the ability to generalise to cross-domain scenarios.

Reference-based instance segmentation Nguyen & Todorovic (2022); Ganea et al. (2021); Han et al. (2022a) offers a promising solution to this challenge by using a small set of annotated reference images to guide the segmentation of a large set of target images. This idea has the potential to enable cheap, quick and automatic annotation of datasets where such labelling is expensive, time-consuming and requires expertise knowledge Benenson et al. (2019); Papadopoulos et al. (2021). Unlike slow manual prompting Hu et al. (2024), using reference images can incorporate semantic understanding directly from examples, and is thus well-suited for automated segmentation tasks. Despite promising results, we observe that existing reference-based segmentation methods often require fine-tuning on novel classes and this raises a set of well understood concerns that include task-specific data requirements, overfitting and domain shift. We conjecture that a prospective alternative approach to guiding reference-based instance segmentation involves reusing the general

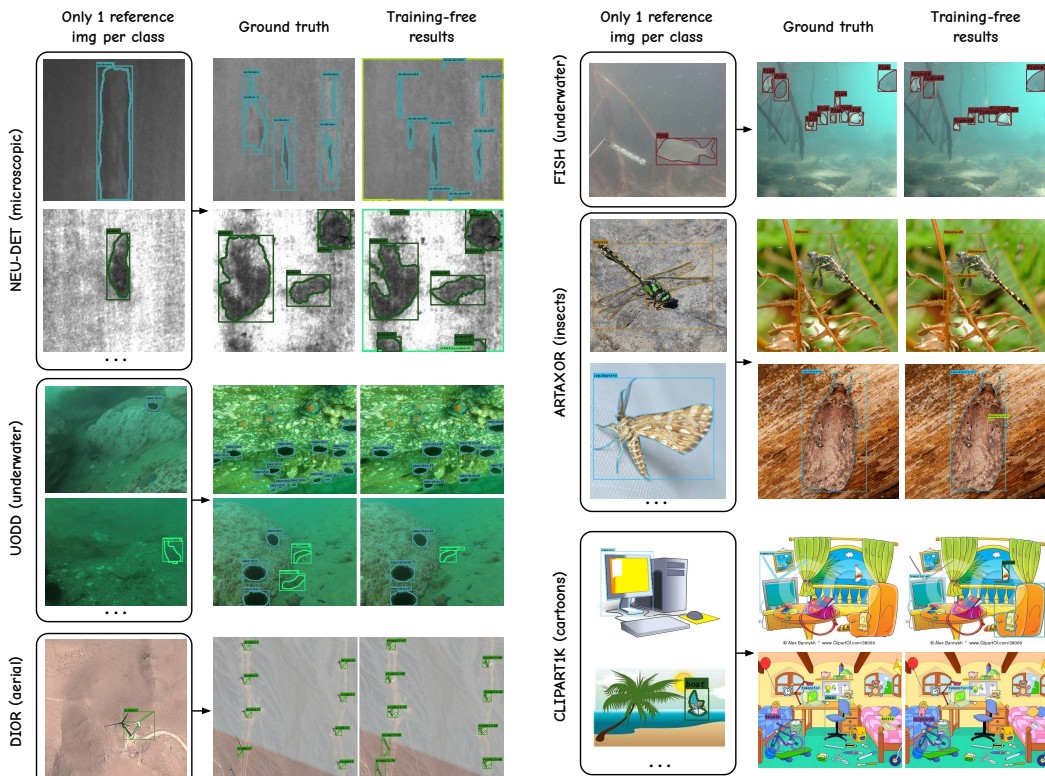

Figure 1: **Cross-domain 1-shot segmentation results using our training-free method on CD-FSOD benchmark**. Our method directly evaluates on diverse datasets without any fine-tuning, using frozen SAMv2 and DINOv2 models. The reference set contains a single example image per class. The model then segments the entire target dataset based on the reference set. Results show: (1) generalization capabilities to out-of-distribution domains (e.g., underwater images, cartoons, microscopic textures); (2) state-of-the-art performance in 1-shot segmentation without training or domain adaptation; (3) limitations in cases with ambiguous annotations or highly similar classes (e.g., "harbor" vs. "ships" in DIOR). Best viewed when zoomed in. Ablation studies further investigate the variance associated with the selection of reference images. See appendix for more visualisation.

purpose capabilities of vision foundation models Caron et al. (2021); Radford et al. (2021); Oquab et al. (2024); Kirillov et al. (2023); Ravi et al. (2024).

Several works Liu et al. (2024c); Sun et al. (2024); Wang et al. (2023c) have attempted to combine pretrained models for reference-based segmentation, e.g. Matcher Liu et al. (2024c) which integrates DINOv2 with SAM for semantic segmentation tasks. However, these methods face several limitations. Firstly, they rely on computationally expensive distance metrics (Earth Mover's Distance), and complex thresholding mechanisms, which significantly slow down inference. Secondly, they are not suited for instance-level segmentation tasks, struggling with fine-grained discrimination in complex multi-object scenes. In fact, the instance segmentation setting presents unique challenges—how do we handle occlusions, scale variations, ambiguous object boundaries and varying image quality, all with just a few reference images—and they should be tackled thoughtfully. Effectively combining foundation models, without significant finetuning, remains a large challenge Espinosa et al. (2024), particularly when attempting to leverage generalisation capabilities of semantic ViT backbones (e.g. DINOv2), to achieve precise localisation Zhang et al. (2024b). These observations highlight that naively composing existing foundation models is insufficient for effective instance-level matching; achieving strong performance requires a dedicated system-level design, which motivates the training-free method we develop in this work.

We propose a training-free three-stage method: (1) constructing a memory bank of category-specific features, (2) refining feature representations via two-step aggregation, and (3) performing inference through feature matching and a novel semantic-aware soft merging strategy. This results in a

training-free, high-performing framework that achieves significant performance gains on established datasets. Furthermore, our approach maintains its effectiveness across diverse domains with fixed hyperparameters, making it accessible for a wide range of applications. Our contributions are:

- We propose a training-free method that effectively integrates semantic-agnostic segmentation mask proposals with fine-grained semantics for reference-based instance segmentation.
- We introduce a novel three-stage framework for instance segmentation with vision foundation models, addressing key integration challenges with (1) memory bank construction, (2) two-step feature aggregation, and (3) feature matching with semantic-aware soft merging.
- Our method achieves state-of-the-art performance on COCO-FSOD, PASCAL-FSOD, and CD-FSOD benchmarks, demonstrating strong generalisation across diverse datasets under fixed hyperparameter settings, without the need for intermediate fine-tuning.

## 2 RELATED WORK

**Reference-based Instance Segmentation** aims to segment individual objects within an image, distinguishing between instances of the same category Chen et al. (2019); Li et al. (2017). Traditional approaches, such as Mask R-CNN He et al. (2017), use region proposals with convolutional networks to predict instance masks, while transformer-based models like DETR Carion et al. (2020) and Mask2Former Cheng et al. (2022) integrate global context through self-attention mechanisms. These methods have demonstrated success in standard instance segmentation tasks by leveraging large, labeled datasets Lin et al. (2014); Gupta et al. (2019). Reference-based instance segmentation extends these tasks to handle novel categories with limited labeled examples. Early works Ganea et al. (2021); Nguyen & Todorovic (2022) adapted Mask R-CNN by introducing instance-level discriminative features Ganea et al. (2021) or uncertainty-guided bounding box prediction Nguyen & Todorovic (2022). More recent works have unified segmentation tasks into in-context learning frameworks Wang et al. (2023b;c); Jiang et al. (2024); Ren et al. (2024), involving an expensive pretraining phase on a wide range of segmentation tasks, including instance segmentation Wang et al. (2023b;c); Ren et al. (2024) and using contrastive pretraining to integrate visual and textual prompts Jiang et al. (2024). Despite these advancements, reference-based instance segmentation remains challenging due to the lack of labeled data, the complexity of multi-instance scenarios, limited generalisation across domains for specialist models, reference image ambiguity, and reliance on predefined class labels Zang et al. (2021). Additionally, reusing frozen backbones not originally pretrained for instance segmentation remains a challenge Zhang et al. (2024b); Espinosa et al. (2024). Our method effectively reuses two existing frozen vision foundation models—none of which was trained for reference-based instance segmentation task— to tackle reference-based instance segmentation without additional training, while also generalising well to unusual domains.

**Vision Foundation models** have revolutionised computer vision by learning strong, pretrained representations, transferable across diverse tasks. CLIP Radford et al. (2021) and DINO models Caron et al. (2021); Oquab et al. (2024) exemplify this trend, using contrastive learning to align visual and textual representations, and learning robust image embeddings from unlabeled data, respectively. These models have been widely adopted for downstream tasks, including open-vocabulary detection Shin et al. (2024); Han et al. (2023) and semantic segmentation Salehi et al. (2023); Ziegler & Asano (2022). However, CLIP struggles with detailed spatial reasoning, while DINOv2, despite capturing fine-grained semantics, produces low-resolution feature maps. The Segment Anything Models (SAM, SAM2) Kirillov et al. (2023); Ravi et al. (2024) are a notable addition to this category, trained on an extensive, category-agnostic dataset (SA-1B) Kirillov et al. (2023). While SAM excels in generating segmentation masks with minimal input (e.g., points, bounding boxes), it lacks inherent semantic understanding Shin et al. (2024); Espinosa et al. (2024); Han et al. (2023); Ji et al. (2024). Efforts to bridge this gap include pairing SAM with language models Lai et al. (2024), diffusion models Zhu et al. (2024), or fine-tuning it on labeled datasets like COCO Lin et al. (2014) and ADE20K Li et al. (2024). However, these adaptations often result in complex pipelines or limited scalability. Furthermore, SAM's semantic-agnostic nature poses limitations in scenarios requiring class differentiation at instance-level. Our work combines the complementary strengths of DINOv2 and SAM without requiring finetuning. By aggregating and matching features from multiple references, we enable high-precision, training-free instance segmentation, achieving state-of-the-art results on diverse few-shot benchmarks.

**Automatic Vision Prompting for SAM** aims to construct automatic prompting pipelines to enhance SAM's versatility in complex visual tasks, reducing its reliance on manual inputs. Training-free methods Zhang et al. (2024a); Liu et al. (2024c) leverage feature-matching techniques but often rely on manually tuned thresholds, distance metrics and complex pipelines. Other approaches focus on learning prompts directly, such as spatial or semantic optimisation Jia et al. (2022); Sun et al. (2024); Huang et al. (2024), but these methods still face challenges in multi-instance or semantically dense settings. Zero-shot methods Yao et al. (2024); Shtedritski et al. (2023) introduce visual markers to guide attention during segmentation, but lack fine-grained precision. Recent efforts, including SEEM Zou* et al. (2023), have unified segmentation and recognition tasks through shared decoders, while SINE Liu et al. (2024b) tackles task ambiguity by disentangling segmentation tasks. Others have paired SAM with Stable Diffusion for open-vocabulary segmentation Zhu et al. (2024), enabling it to incorporate semantic cues. Similarly, LISA Lai et al. (2024) utilises language-based instructions to adapt SAM for text-guided tasks. Despite these efforts, handling multi-instance scenarios and semantic ambiguity without training remains challenging. Our method differs by integrating SAM with DINOv2 without additional training or prompt optimisation. We introduce a multi-stage method (memory bank construction, representation aggregation, and semantic-aware feature matching) with fixed hyperparameters across all experimental settings, making it practical for a wide range of downstream applications and directly accessible to practitioners across domains.

## 3 METHOD

### 3.1 PRELIMINARIES

**Segment Anything Model (SAM)** Kirillov et al. (2023) is designed for promptable segmentation, that is, it responds to different types of geometric prompts to generate image segmentation masks. It consists of three main components: an image encoder, a prompt encoder, and a mask decoder. The image encoder is a pretrained Vision Transformer (ViT) Dosovitskiy et al. (2021) adapted for high-resolution inputs Li et al. (2022a). The prompt encoder handles both sparse (points, boxes, text) and dense (rough mask) prompts, which are encoded with positional encodings Tancik et al. (2020), learnt embeddings, and off-the-shelf text encoders. The mask decoder efficiently generates masks by using a modified Transformer decoder block Carion et al. (2020) with self-attention and cross-attention on the prompts. For training, SAM uses a combination of focal loss Lin et al. (2017) and dice loss Milletari et al. (2016).

**DINOv2 Self-supervised pretrained vision encoder** Oquab et al. (2024) is a self-supervised vision model designed to produce general-purpose visual features. It uses a discriminative self-supervised learning approach based on Vision Transformer (ViT) architectures Dosovitskiy et al. (2021). DINOv2 makes use of teacher-student networks, incorporates Sinkhorn-Knopp normalisation, multi-crops strategy, separate projection heads for image-level and patch-level objectives, and additional regularisation techniques to stabilise and scale training Caron et al. (2021); Zhou et al. (2022a). It is trained on a curated dataset (LVD-142M) of 142 million images using efficient training techniques (fast memory-efficient attention, stochastic depth, etc.). Its features are transferable across tasks and domains, making it a robust backbone for both global and local visual understanding tasks.

**Reference-based instance segmentation** aims to segment a target image by using a reference segmented image as an example (rather than geometric prompts). More formally, given a reference image $I_r$ and its corresponding annotations $M_r^i$ (where $i$ denotes different object categories), we use this data to segment the corresponding regions in the target image $I_t$ that also belong to category $i$.

### 3.2 TRAINING-FREE METHOD

The goal of our training-free method is to extract category-specific features from a set of annotated reference examples and use them to segment and classify instances in target images. Unlike methods that require model retraining, we employ a memory-based approach to store discriminative representations for object categories. It consists of three main stages: (1) *constructing a memory bank* from reference images, (2) refining these representations via *two-stage feature aggregation*, and (3) performing *inference on the target images* through feature matching and semantic-aware soft merging. Figure 2 illustrates the complete pipeline.

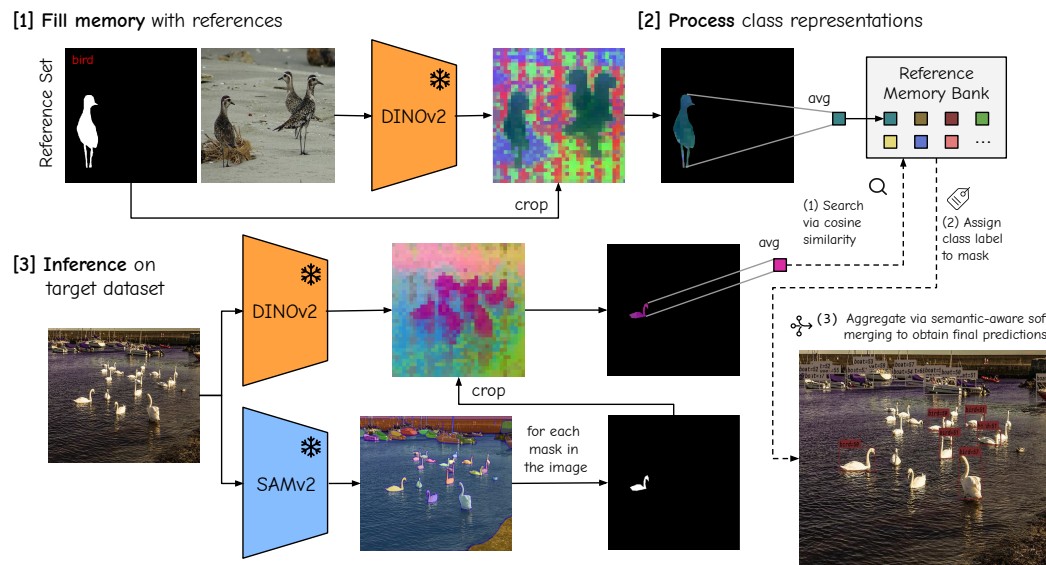

Figure 2: **Overview of our training-free method for few-shot instance segmentation and object detection.** (1) *Reference Memory Creation*: A segmented reference image is processed using the DINOv2 model to generate semantic feature embeddings. (2) *Feature aggregation*: We compute instance-wise feature representations, and then, aggregate them into class-wise prototypes, stored in the memory bank. (3) *Inference on Target Dataset*: For each target image, SAMv2 generates instance segmentation masks while DINOv2 extracts semantic features. Using cosine similarity, each mask's embedding is compared with the reference memory bank to assign the most similar class label. Finally, predictions are aggregated via semantic-aware soft merging to produce the final annotated image. This pipeline enables semantic prompting via reference images, without requiring fine-tuning, and demonstrates state-of-the-art performance on established benchmarks (COCO-FSOD, PASCAL-FSOD) and strong generalization across domains (CD-FSOD).

**(1) Memory Bank Construction.** Given a set of reference images $\{I_r^j\}_{j=1}^{N_r}$ and their corresponding instance masks $\{M_r^{j,i}\}_{j=1}^{N_r}$ for category $i$, we extract dense feature maps $F_r^j \in \mathbb{R}^{H' \times W' \times d}$ using a pretrained frozen encoder $\mathcal{E}^1$, where $d$ is the feature dimension and $H', W'$ denote the spatial resolution of the feature map. The corresponding instance masks $M_r^{j,i} \in \{0,1\}^{H' \times W'}$ are resized to match this resolution. For each category $i$, we store the masked features $\mathcal{F}_r^{j,i} = F_r^j \odot M_r^{j,i}$ where $\odot$ denotes element-wise multiplication. These category-wise feature sets are stored in a memory bank $\mathcal{M}_i$, which is synchronised across GPUs to ensure consistency in distributed settings.

**(2) Two-stage feature aggregation.** To construct category prototypes, we first compute instance-wise feature representations, and then, aggregate them into class-wise prototypes.

(a) **Instance-wise prototypes:** Each instance $k$ in reference image $I_r^j$ has its own prototype, computed by averaging the feature embeddings within its corresponding mask: $P_r^{j,k} = \frac{1}{\|M_r^{j,k}\|_1} \sum_{(u,v)} M_r^{j,k}(u,v) F_r^j(u,v)$ where $P_r^{j,k} \in \mathbb{R}^d$ is the mean feature representation of the $k$-th instance in image $I_r^j$.

(b) **Class-wise prototype:** We compute the category prototype $P_i$ by averaging all instance-wise prototypes belonging to the same category $i$: $P_i = \frac{1}{N_i} \sum_{j=1}^{N_r} \sum_{k \in \mathcal{K}_i^j} P_r^{j,k}$, where $\mathcal{K}_i^j$ is the set of instances in image $I_r^j$ that belong to category $i$, and $N_i = \sum_{j=1}^{N_r} |\mathcal{K}_i^j|$ is the total number of instances belonging to category $i$. These class-wise prototypes $P_i$ are stored in the memory bank.

---

[1] https://github.com/facebookresearch/dinov2

**(3) Inference on Target Images.** For a target image $I_t$, we extract dense features $F_t \in \mathbb{R}^{H' \times W' \times d}$ using the same encoder $\mathcal{E}$. We use the frozen SAM model to generate $N_m$ candidate instance masks $\{M_t^m\}_{m=1}^{N_m}$, where $M_t^m \in \{0,1\}^{H' \times W'}$. Each mask $M_t^m$ is used to compute a feature representation $P_t^m$ via average pooling and L2 normalisation: $P_t^m = \frac{1}{\|M_t^m\|_1} \sum_{(u,v)} M_t^m(u,v) F_t(u,v)$, $\quad \hat{P}_t^m = \frac{P_t^m}{\|P_t^m\|_2}$ where $\hat{P}_t^m \in \mathbb{R}^d$

To classify each candidate mask, we compute:

(a) **Feature Matching.** We compute the cosine similarity between $\hat{P}_t^m$ and category prototypes $P_i$, which provides the classification score $S_t^m$ for mask $M_t^m$: $S_t^m = \max_i \left( \frac{\hat{P}_t^m \cdot P_i}{\|P_i\|_2} \right)$

(b) **Semantic-Aware Soft Merging.** To handle overlapping predictions, we introduce a novel soft merging strategy. Given two masks $M_t^m$ and $M_t^{m'}$ of the same category, we compute their intersection-over-self (IoS), and weight it by feature similarity.

$$\text{IoS}(M_t^m, M_t^{m'}) = \frac{\sum(M_t^m \cap M_t^{m'})}{\sum M_t^m} \qquad w_{m,m'} = \frac{\hat{P}_t^m \cdot \hat{P}_t^{m'}}{\|\hat{P}_t^{m'}\|_2}$$

The final score for each mask is adjusted using a decay factor: $S_t^m \leftarrow S_t^m \cdot \sqrt{(1 - \text{IoS}(M_t^m, M_t^{m'})w_{m,m'})}$ reducing redundant detections while preserving distinct instances that may partially overlap. We rank masks by their adjusted scores, and select top-$K$ predictions as final output.

### 3.3 Technical implementation details

Our codebase builds upon SAM2-L (Hierarchical ViT) for mask generation and DINOv2-L as the feature encoder. The encoder processes images at 518×518 resolution with a patch size of 14×14, while SAM2 operates at 1024×1024 resolution. During inference, SAM2 first generates candidate masks using a 32×32 grid of query points. For each mask, we compute L2-normalised features by average pooling the encoder features within the masked region. These features are compared against our memory bank, which stores features from $n$ reference images per category. We employ non-maximum suppression with an IoU threshold of 0.5, followed by our semantic-aware soft merging strategy to handle overlapping predictions. The model outputs up to 100 instances per image. The implementation uses PyTorch Ansel et al. (2024) and PyTorch Lightning Falcon & The PyTorch Lightning team (2019) for distributed workload across GPUs.

## 4 Results

### 4.1 Object Detection and Instance Segmentation

Although our training-free method outputs segmentation masks, we convert instance masks to bounding boxes for a fair comparison with existing methods.

**COCO-FSOD Benchmark.** We evaluate our method in a strict few-shot setting on the COCO-$20^i$ dataset Lin et al. (2014); Kang et al. (2019a), using the standard 10-shot and 30-shot settings. Results for the COCO-FSOD benchmark are shown in Table 1. All results are reported for COCO-NOVEL classes. Novel classes are the COCO categories that intersect with PASCAL VOC categories Everingham et al. (2010). Our method achieves state-of-the-art while being completely training-free, outperforming approaches that fine-tune on novel classes. Figure 3 presents qualitative results, showing our method's ability to handle multiple overlapping instances in crowded scenes with fine-grained semantics and precise localisation. With semantic-aware soft merging, we mitigate duplicate detections and false positives. Failure cases are discussed in the Appendix.

**PASCAL VOC Few-Shot Benchmark.** The PASCAL-VOC dataset Everingham et al. (2010) consists of 20 classes. For few-shot evaluation, we adopt the standard approach Guirguis et al. (2023), splitting the classes into three groups, each with 15 base and 5 novel classes. As in prior work Zhang et al. (2024b), we report AP50 results on the novel classes. Table 2 shows that our method outperforms all previous approaches across all splits, achieving state-of-the-art performance across all splits. This holds for both methods that fine-tune on novel classes and those that do not.

| Method | Ft. on novel | 10-shot | | | 30-shot | | |
|---|---|---|---|---|---|---|---|
| | | nAP | nAP50 | nAP75 | nAP | nAP50 | nAP75 |
| TFA Wang et al. (2020) | ✓ | 10.0 | 19.2 | 9.2 | 13.5 | 24.9 | 13.2 |
| FSCE Sun et al. (2021) | ✓ | 11.9 | – | 10.5 | 16.4 | – | 16.2 |
| Retentive RCNN Fan et al. (2021) | ✓ | 10.5 | 19.5 | 9.3 | 13.8 | 22.9 | 13.8 |
| HeteroGraph Han et al. (2021) | ✓ | 11.6 | 23.9 | 9.8 | 16.5 | 31.9 | 15.5 |
| Meta F. R-CNN Han et al. (2022a) | ✓ | 12.7 | 25.7 | 10.8 | 16.6 | 31.8 | 15.8 |
| LVC Kaul et al. (2022) | ✓ | 19.0 | 34.1 | 19.0 | 26.8 | 45.8 | 27.5 |
| C. Transformer Han et al. (2022b) | ✓ | 17.1 | 30.2 | 17.0 | 21.4 | 35.5 | 22.1 |
| NIFF Guirguis et al. (2023) | ✓ | 18.8 | – | – | 20.9 | – | – |
| DiGeo Ma et al. (2023) | ✓ | 10.3 | 18.7 | 9.9 | 14.2 | 26.2 | 14.8 |
| CD-ViTO (ViT-L) Fu et al. (2025) | ✓ | 35.3 | 54.9 | 37.2 | 35.9 | 54.5 | 38.0 |
| FSRW Kang et al. (2019b) | ✗ | 5.6 | 12.3 | 4.6 | 9.1 | 19.0 | 7.6 |
| Meta R-CNN Yan et al. (2019) | ✗ | 6.1 | 19.1 | 6.6 | 9.9 | 25.3 | 10.8 |
| DE-ViT (ViT-L) Zhang et al. (2024b) | ✗ | 34.0 | 53.0 | 37.0 | 34.0 | 52.9 | 37.2 |
| **Training-free (ours)** | ✗ | **36.6** | **54.1** | **38.3** | **36.8** | **54.5** | **38.7** |

Table 1: Comparison of our training-free method against state-of-the-art approaches on the COCO-FSOD benchmark under 10-shot and 30-shot settings. Our approach achieves state-of-the-art performance without finetuning on novel classes (Ft. on novel). Results are reported in terms of nAP, nAP50, and nAP75. nAP refers to mAP for novel classes. Competing methods results are sourced from Fu et al. (2025). Since we are the only method that provides both bounding box and segmentation results, for simplicity we omit segmentation AP on this table.

| Method | Ft. on novel | Novel Split 1 | | | | | Novel Split 2 | | | | | Novel Split 3 | | | | | Avg |
|---|---|---|---|---|---|---|---|---|---|---|---|---|---|---|---|---|---|
| | | 1 | 2 | 3 | 5 | 10 | 1 | 2 | 3 | 5 | 10 | 1 | 2 | 3 | 5 | 10 | |
| FsDetView Xiao & Marlet (2020) | ✓ | 25.4 | 20.4 | 37.4 | 36.1 | 42.3 | 22.9 | 21.7 | 22.6 | 25.6 | 29.2 | 32.4 | 19.0 | 29.8 | 33.2 | 39.8 | 29.2 |
| TFA Wang et al. (2020) | ✓ | 39.8 | 36.1 | 44.7 | 55.7 | 56.0 | 23.5 | 26.9 | 34.1 | 35.1 | 39.1 | 30.8 | 34.8 | 42.8 | 49.5 | 49.8 | 39.9 |
| Retentive RCNN Fan et al. (2021) | ✓ | 42.4 | 45.8 | 45.9 | 53.7 | 56.1 | 21.7 | 27.8 | 35.2 | 37.0 | 40.3 | 30.2 | 37.6 | 43.0 | 49.7 | 50.1 | 41.1 |
| DiGeo Ma et al. (2023) | ✓ | 37.9 | 39.4 | 48.5 | 58.6 | 61.5 | 26.6 | 28.9 | 41.9 | 42.1 | 49.1 | 30.4 | 40.1 | 46.9 | 52.7 | 54.7 | 44.0 |
| HeteroGraph Han et al. (2021) | ✓ | 42.4 | 51.9 | 55.7 | 62.6 | 63.4 | 25.9 | 37.8 | 46.6 | 48.9 | 51.1 | 35.2 | 42.9 | 47.8 | 54.8 | 53.5 | 48.0 |
| Meta Faster R-CNN Han et al. (2022a) | ✓ | 43.0 | 54.5 | 60.6 | 66.1 | 65.4 | 27.7 | 35.5 | 46.1 | 47.8 | 51.4 | 40.6 | 46.4 | 53.4 | 59.9 | 58.6 | 50.5 |
| CrossTransformer Han et al. (2022b) | ✓ | 49.9 | 57.1 | 57.9 | 63.2 | 67.1 | 27.6 | 34.5 | 43.7 | 49.2 | 51.2 | 39.5 | 54.7 | 52.3 | 57.0 | 58.7 | 50.9 |
| LVC Kaul et al. (2022) | ✓ | 54.5 | 53.2 | 58.8 | 63.2 | 65.7 | 32.8 | 29.2 | 50.7 | 49.8 | 50.6 | 48.4 | 52.7 | 55.0 | 59.6 | 59.6 | 52.3 |
| NIFF Guirguis et al. (2023) (*) | ✓ | 62.8 | 67.2 | 68.0 | 70.3 | 68.8 | 38.4 | 42.9 | 54.0 | 56.4 | 54.0 | 56.4 | 62.1 | 61.2 | 64.1 | 63.9 | 59.4 |
| Multi-Relation Det Fan et al. (2020) | ✗ | 37.8 | 43.6 | 51.6 | 56.5 | 58.6 | 22.5 | 30.6 | 40.7 | 43.1 | 47.6 | 31.0 | 37.9 | 43.7 | 51.3 | 49.8 | 43.1 |
| DE-ViT (ViT-S/14) Zhang et al. (2024b) | ✗ | 47.5 | 64.5 | 57.0 | 68.5 | 67.3 | 43.1 | 34.1 | 49.7 | 56.7 | 60.8 | 52.5 | 62.1 | 60.7 | 61.4 | 64.5 | 56.7 |
| DE-ViT (ViT-B/14) Zhang et al. (2024b) | ✗ | 56.9 | 61.8 | 68.0 | 73.9 | 72.8 | 45.3 | 47.3 | 58.2 | 59.8 | 60.6 | 58.6 | 62.3 | 62.7 | 64.6 | 67.8 | 61.4 |
| DE-ViT (ViT-L/14) Zhang et al. (2024b) | ✗ | 55.4 | 56.1 | 68.1 | 70.9 | 71.9 | 43.0 | 39.3 | 58.1 | 61.6 | 63.1 | 58.2 | 64.0 | 61.3 | 64.2 | 67.3 | 60.2 |
| **Training-free (ours)** | ✗ | **70.8** | **72.3** | **73.3** | **77.2** | **79.1** | **54.5** | **67.0** | **76.3** | **75.9** | **78.2** | **61.1** | **67.9** | **71.3** | **70.8** | **72.6** | **71.2** |

Table 2: AP50 results on the novel classes of the Pascal VOC few-shot benchmark. Competing method results are sourced from Zhang et al. (2024b). State-of-the-art results are highlighted in **bold**. (*) indicates that the corresponding implementation is not publicly accessible. Our proposed training-free approach consistently achieves superior performance across all splits, outperforming fine-tuned methods.

## 4.2 CROSS-DOMAIN FEW-SHOT OBJECT DETECTION

The CD-FSOD benchmark (Fu et al., 2025) is designed to evaluate cross-domain few-shot object detection (CD-FSOD) models by addressing challenges in domain shifts and limited data scenarios. It uses COCO as the source training dataset (SD), and six target datasets (TD) — ArTaxOr, Clipart1k, DIOR, DeepFish, NEUDET, and UODD — spanning photorealistic, cartoon, aerial, underwater, and industrial domains with high inter-class variance. While many approaches fine-tune on a few labeled instances (support set S) from TD before testing on the query set Q, our model is entirely *training-free*. Thus, we directly evaluate on the six target datasets without any fine-tuning.

Table 3 compares FSOD methods on the CD-FSOD benchmark across 1-shot, 5-shot, and 10-shot settings. Our method sets a new state-of-the-art among training-free approaches and remains competitive with fine-tuned models. These results demonstrate its strong cross-domain generalisation and robustness without requiring retraining.

## 4.3 FEW-SHOT SEMANTIC SEGMENTATION ON COCO

Although our method is designed for instance segmentation, we also evaluate it on the COCO-20$^i$ Few-Shot Semantic Segmentation benchmark Nguyen & Todorovic (2019). The 80 COCO classes

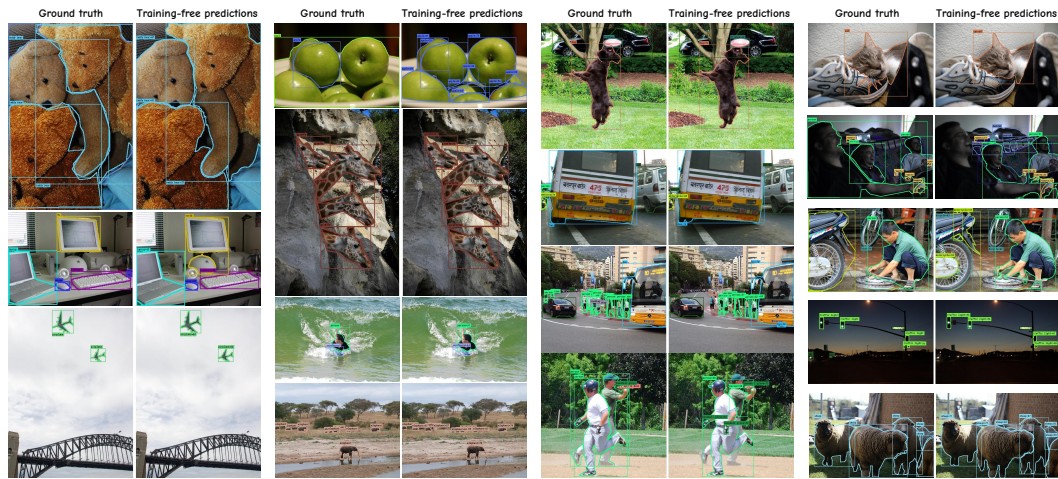

Figure 3: Qualitative results on the COCO val2017 test set under the 10-shot setting (using 10 reference images per class). Bounding box visualisations are thresholded at 0.5. Our method effectively handles multiple overlapping instances in crowded scenes, demonstrating fine-grained semantics and precise localisation. Through semantic-aware soft merging, we avoid duplicate detections and false positives. Best viewed when zoomed in.

| Method | Ft. on novel | ArT axOr | Clip art1k | DIOR | Deep Fish | NEU DET | UO DD | Avg |
|---|---|---|---|---|---|---|---|---|
| **Fine-tuned on novel classes** | | | | | | | | |
| TFA w/cos ○ Wang et al. (2020) | ✓ | 3.1/8.8/14.8 | -/-/- | 8.0/18.1/20.5 | -/-/- | -/-/- | 4.4/8.7/11.8 | -/-/- |
| FSCE ○ Sun et al. (2021) | ✓ | 3.7/10.2/15.9 | -/-/- | 8.6/18.7/21.9 | -/-/- | -/-/- | 3.9/9.6/12.0 | -/-/- |
| DeFRCN ○ Qiao et al. (2021) | ✓ | 3.6/9.9/15.5 | -/-/- | 9.3/18.9/22.9 | -/-/- | -/-/- | 4.5/9.9/12.1 | -/-/- |
| Distill-cdfsd ○ Xiong (2023) | ✓ | 5.1/12.5/18.1 | 7.6/23.3/27.3 | 10.5/19.1/26.5 | nan/15.5/15.5 | nan/16.0/21.1 | 5.9/12.2/14.5 | -/16.4/20.5 |
| ViTDeT-FT† Li et al. (2022b) | ✓ | 5.9/20.9/23.4 | 6.1/23.3/25.6 | 12.9/23.3/29.4 | 0.9/9.0/6.5 | 2.4/13.5/15.8 | 4.0/11.1/15.6 | 5.4/16.9/19.4 |
| Detic-FT† Zhou et al. (2022b) | ✓ | 3.2/8.7/12.0 | 15.1/20.2/22.3 | 4.1/12.1/15.4 | 9.0/14.3/17.9 | 3.8/14.1/16.8 | 4.2/10.4/14.4 | 6.6/13.3/16.5 |
| DE-ViT-FT† Zhang et al. (2024b) | ✓ | 10.5/38.0/49.2 | 13.0/38.1/40.8 | 14.7/23.4/25.6 | 19.3/21.2/21.3 | 0.6/7.8/8.8 | 2.4/5.0/5.4 | 10.1/22.3/25.2 |
| CD-ViTO† Fu et al. (2025) | ✓ | 21.0/47.9/60.5 | 17.7/41.1/44.3 | 17.8/26.9/30.8 | 20.3/22.3/22.3 | 3.6/11.4/12.8 | 3.1/6.8/7.0 | 13.9/26.1/29.6 |
| **Training-free (no novel class fine-tuning)** | | | | | | | | |
| Meta-RCNN ○ Yan et al. (2019) | ✗ | 2.8/8.5/14.0 | -/-/- | 7.8/17.7/**20.6** | -/-/- | -/-/- | 3.6/8.8/11.2 | -/-/- |
| Detic† Zhou et al. (2022b) | ✗ | 0.6/0.6/0.6 | 11.4/11.4/11.4 | 0.1/0.1/0.1 | 0.9/0.9/0.9 | 0.0/0.0/0.0 | 0.0/0.0/0.0 | 2.2/2.2/2.2 |
| DE-ViT† Zhang et al. (2024b) | ✗ | 0.4/10.1/9.2 | 0.5/5.5/11.0 | 2.7/7.8/8.4 | 0.4/2.5/2.1 | 0.4/1.5/1.8 | 1.5/3.1/3.1 | 1.0/5.1/5.9 |
| **Training-free (ours)** | ✗ | **28.2/35.7/35.0** | **18.9/24.9/25.9** | **14.9/18.5**/16.4 | **30.5/29.6/29.6** | **5.5/5.2/5.5** | **10.0/20.2/16.0** | **18.0/22.4/21.4** |

Table 3: Per-cell values are 1-shot/5-shot/10-shot mAP on the CD-FSOD benchmark. The ○ symbol indicates results sourced from Distill-cdfsod (Xiong, 2023), while † denotes results reported by CD-ViTO (Fu et al., 2025). 'Avg.' represents the average performance across datasets.

are divided into four folds Hu et al. (2019); Wang et al. (2019); Nguyen & Todorovic (2019), each containing 60 base and 20 novel classes. We assess performance on the 20 novel classes under strict 1-shot and 5-shot settings Zhu et al. (2024). Results are shown in Table 4. To adapt our instance segmentation predictions to semantic segmentation, we aggregate all instances of the same class into semantic maps, enabling direct comparison with prior methods. Despite being entirely *training-free*, our method achieves competitive performance against fine-tuned approaches.

## 4.4 ABLATIONS

**Variance in reference set.** Using different reference images leads to variations in results, as performance depends on the quality of the selected reference images for each class. To quantify this variance, we evaluate our method on the COCO-$20^i$ few-shot object detection benchmark using different random seeds to select reference images. Figure 5 shows the standard deviation (std) across 10 runs. We observe that increasing the number of reference images (higher n-shots) reduces result variance, with lower std values. In 1, 2, and 3-shot settings, reference image selection has a more noticeable impact, while for 5 or more shots, the low std demonstrates robustness to reference set variation. This suggests that some reference images are inherently stronger for a given shot.

| Methods | FT | 1-shot | | | | | 5-shot | | | | |
|---|---|---|---|---|---|---|---|---|---|---|---|
| | | $20^0$ | $20^1$ | $20^2$ | $20^3$ | mean | $20^0$ | $20^1$ | $20^2$ | $20^3$ | mean |
| DiffewS Zhu et al. (2024) | ✓ | 47.7 | 56.4 | 51.9 | 48.7 | 51.2 | 52.0 | 63.0 | 54.5 | 54.3 | 56.0 |
| DiffewS-n Zhu et al. (2024) | ✓ | 47.1 | 56.6 | 53.8 | 48.3 | 52.2 | 57.3 | 66.5 | 60.3 | 58.8 | 60.7 |
| HSNet Min et al. (2021) | ✗ | 37.2 | 44.1 | 42.4 | 41.3 | 41.2 | 45.9 | 53.0 | 51.8 | 47.1 | 49.5 |
| CyCTR Zhang et al. (2021) | ✗ | 38.9 | 43.0 | 39.6 | 39.8 | 40.3 | 41.1 | 48.9 | 45.2 | 47.0 | 45.6 |
| VAT Hong et al. (2022) | ✗ | 39.0 | 43.8 | 42.6 | 39.7 | 41.3 | 44.1 | 51.1 | 50.2 | 46.1 | 47.9 |
| BAM Lang et al. (2022) | ✗ | 43.4 | 50.6 | 47.5 | 43.4 | 46.2 | 49.3 | 54.2 | 51.6 | 49.6 | 51.2 |
| DCAMA Shi et al. (2022) | ✗ | 49.5 | 52.7 | 52.8 | 48.7 | 50.9 | 55.4 | 60.3 | 59.9 | 57.5 | 58.3 |
| HDMNet Peng et al. (2023) | ✗ | 43.8 | 55.3 | 51.6 | 49.4 | 50.0 | 50.6 | 61.6 | 55.7 | 56.0 | 56.0 |
| **Training-free** | ✗ | 32.1 | 46.6 | 50.9 | 48.7 | 44.6 | 55.2 | 49.5 | 60.7 | 45.5 | 52.7 |

Table 4: Performance comparison of strict few-shot semantic segmentation settings (1 and 5-shot) on COCO-$20^i$. We aggregate instance-level predictions to allow comparison with semantic segmentation works. Previous methods results are sourced from Zhu et al. (2024). FT refers to finetuning on novel classes.

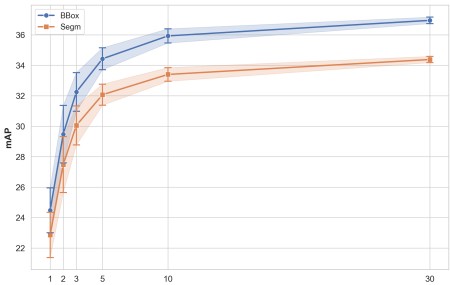

Table 5: Variance (mAP) n-shot on COCO-$20^i$. Error bars show std over 10 runs with different reference image sets. Higher variance at low n-shot reflects sensitivity to reference images; it decreases as n increases, demonstrating our method's robustness.

**Aggregation Strategies.** The improvements of our soft-merging semantic-aware strategy are shown in Tab. 6. Other aggregation variants, such as covariance similarity, instance softmax, score decay, iterative mask refine, attention-guided global average, underperformed. Earth Mover's Distance (EMD) was investigated in early design, but we found that its optimal-transport formulation introduced substantial computational overhead without providing clear qualitative benefits over cosine similarity.

**Inference Runtime Efficiency.** Our training-free stages are optimised and lightweight. (1) Memory bank construction (0.1 s/img) is computed once, only requiring to encode $n$ reference images per class with DINOv2. Reference-image features are pre-cached for all following steps. (2) Semantic matching (0.0003 s/img) is a fully parallelised dot product of cosine similarities. (3) Soft-merging (0.006 s/img) uses a parallel implementation of NMS. Tab. 7 shows our method yields significant performance gains compared to Matcher Liu et al. (2024c) and speeds up SAM default automatic mask generator (AMG) by $\times 3$ via efficient point sampling, faster mask filtering, and removal of unnecessary post-processing.

| Aggregation strategy | 10-shot nAP |
|---|---|
| Hard-merging (hard threshold of 1 IoS) | 31.2 |
| Soft-merging (without semantics) | 35.7 |
| **Soft-merging (with semantics)** | **36.6** |

Table 6: Ablation on different aggregation and matching strategies.

| Method | Time (sec/img) |
|---|---|
| Matcher Liu et al. (2024c) | 120.014 |
| Training-free (ours) with SAM AMG | 3.5092 |
| **Training-free (ours)** | **0.9292** |

Table 7: Time to process an image on 20 reference classes with an A100 GPU.

**Model Size and Memory Usage.** We report the runtime footprint of our pipeline when using 20 classes and 10 shots per class (200 reference images). At inference time, the combined model size is 1.97 GiB, consisting of the DINOv2 encoder (1.13 GiB) and the SAM2 predictor (856 MiB). CUDA-allocated memory reaches a peak of 11.5 GiB. These measurements provide a clear reference for practical deployment and indicate potential avenues for integrating lighter variants (e.g., MobileSAM Zhang et al. (2023) or LightSAM Cheng et al. (2025)) when targeting edge devices.

## 5 CONCLUSION

In this work, we introduce a novel training-free approach for few-shot instance segmentation by integrating SAM's mask generation capabilities with the fine-grained semantic understanding of DINOv2. Our method uses reference images to construct a memory bank, refines its internal representations with feature aggregation, and performs feature matching for novel instances using cosine similarity and semantic-aware soft merging. We demonstrate that careful engineering of existing frozen

foundation models can lead to state-of-the-art performance without the need for additional training: we achieve 36.8% nAP on COCO-FSOD (outperforming fine-tuned methods), 71.2% nAP50 on PASCAL VOC Few-Shot, and strong generalisation across domains (CD-FSOD benchmark). Furthermore, our semantic segmentation results show that our approach can be extended beyond instance segmentation by aggregating instance predictions into semantic maps.

For future work, we identify several promising directions: (1) exploring learning-based strategies to automatically find the most informative reference images for 1-5 shot scenarios; (2) addressing DINOv2's global semantic biases by improving feature localisation, especially for fine-grained tasks; and (3) investigating lightweight finetuning approaches to improve the internal memory bank representations for 1-5 shot scenarios.

## LLM USAGE

Within the scope of this paper, LLMs are used only to aid and polish writing.

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

# A    APPENDIX

## A.1    MODEL BACKBONE ABLATION

To evaluate the transferability of our method across foundation models, we replace both the semantic encoder (DINOv2) and the SAM-based segmenter with several alternatives. We test CLIP Radford et al. (2021), DINOv3 Siméoni et al. (2025), and PE-Spatial Bolya et al. (2025) models as semantic backbones, and different SAM variants as segmentation backbones. Results are reported in Table 8.

| Semantic backbone | Segm. backb. | 10-shot | | | | | | 30-shot | | | | | |
| | | bbox | | | segm | | | bbox | | | segm | | |
| | | nAP | nAP50 | nAP75 | nAP | nAP50 | nAP75 | nAP | nAP50 | nAP75 | nAP | nAP50 | nAP75 |
|---|---|---|---|---|---|---|---|---|---|---|---|---|---|
| DINOv2-ViT-L-14 | SAM2-L | 35.7 | 52.6 | 37.6 | 33.3 | 52.5 | 35.8 | 36.8 | 54.5 | 38.7 | 34.2 | 54.4 | 36.7 |
| DINOv3-ViT-B-16 | SAM2-L | 32.8 | 47.3 | 34.6 | 30.9 | 47.4 | 33.8 | 33.5 | 48.6 | 35.3 | 31.6 | 48.7 | 34.4 |
| DINOv3-ViT-L-16 | SAM2-L | 33.8 | 48.9 | 35.6 | 32.2 | 49.4 | 35.1 | 34.4 | 50.0 | 36.2 | 32.8 | 50.4 | 35.6 |
| DINOv3-ViT-H-16+ | SAM2-L | 25.5 | 37.0 | 26.6 | 24.4 | 37.3 | 26.6 | 26.2 | 37.9 | 27.2 | 25.0 | 38.2 | 27.4 |
| DINOv3-ViT-H-16+ (px) | SAM2-L | 26.8 | 39.1 | 27.9 | 25.5 | 39.4 | 27.8 | 27.6 | 40.2 | 28.6 | 26.3 | 40.4 | 28.7 |
| CLIP-ViT-B-32 | SAM2-L | 15.2 | 20.4 | 15.8 | 13.9 | 19.8 | 15.2 | 15.8 | 21.2 | 16.6 | 14.6 | 20.7 | 16.2 |
| CLIP-ViT-B-16 | SAM2-L | 19.2 | 26.0 | 20.0 | 18.1 | 25.6 | 20.0 | 19.5 | 26.4 | 20.3 | 18.3 | 26.1 | 20.3 |
| CLIP-ViT-L-14 | SAM2-L | 18.6 | 25.3 | 19.5 | 17.3 | 24.8 | 19.2 | 18.7 | 25.3 | 19.6 | 17.4 | 24.8 | 19.4 |
| CLIP-ViT-L-14-336px | SAM2-L | 17.8 | 24.0 | 18.7 | 16.7 | 23.8 | 18.6 | 17.7 | 23.9 | 18.6 | 16.7 | 23.6 | 18.8 |
| PE-Spatial-L-14-448 (PE) | SAM2-L | 25.7 | 36.3 | 27.1 | 24.1 | 36.6 | 26.3 | 26.5 | 37.4 | 27.9 | 24.7 | 37.5 | 27.1 |
| PE-Spatial-L-14-448 (IN) | SAM2-L | 26.5 | 37.4 | 27.8 | 24.7 | 37.6 | 27.0 | 27.3 | 38.6 | 28.7 | 25.4 | 38.6 | 27.8 |
| PE-Spatial-G-14-448 (PE) | SAM2-L | 24.7 | 34.7 | 26.1 | 23.1 | 34.8 | 25.3 | 24.9 | 35.2 | 26.3 | 23.3 | 35.2 | 25.6 |
| DINOv2-ViT-L-14 | SAM2-T | 27.0 | 43.4 | 26.9 | 26.1 | 43.0 | 27.1 | 27.9 | 45.0 | 27.7 | 26.9 | 44.6 | 27.6 |
| DINOv2-ViT-L-14 | SAM2-S | 29.8 | 45.5 | 31.0 | 27.7 | 44.8 | 29.5 | 30.6 | 47.1 | 31.8 | 28.5 | 46.4 | 30.1 |
| DINOv2-ViT-L-14 | SAM2-B+ | 29.6 | 44.6 | 31.2 | 28.4 | 44.5 | 30.5 | 30.4 | 46.2 | 31.9 | 29.2 | 46.2 | 31.3 |

Table 8: Backbone ablation on COCO-FSOD. We compare different semantic encoders (DINOv2, DINOv3, CLIP, PE-Spatial) paired with the same SAM-based (hiera) segmenter. Similarly, we compare different SAM model sizes paired with the same semantic backbone. Results are reported for 10-shot and 30-shot settings using nAP, nAP50, and nAP75. All experiments use a single seed (33) and no hyperparameter tuning (default prompting and aggregation settings). Results show that the pipeline remains functional across encoder choices, with performance largely influenced by feature-map resolution. PE refers to model's default normalisation, and IN indicates ImageNet normalisation. Model sizes are tiny (T), base (B), large (L), huge (H). Image pixels (px) indicates a larger input image size. 14, 16, 32 refer to the model's patch size.

Overall, our pipeline transfers across backbones with moderate performance variation. CLIP models exhibit a significant performance drop, primarily due to their low-resolution and noisier feature maps. In contrast, PE-Spatial and DINOv3 achieve competitive results with only modest degradation. Since all models were used off-the-shelf without hyperparameter adjustments, we expect these backbones to benefit from more careful integration.

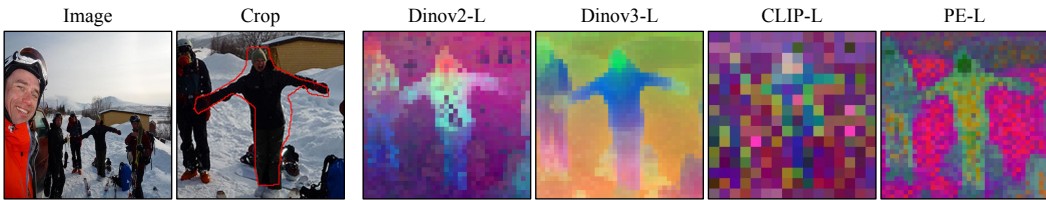

Figure 4: *Feature comparison across semantic backbones.* We show the full image, its cropped region of interest, and the PCA projection of the feature maps from DINOv2, DINOv3, CLIP, and PE-Spatial. CLIP features are low-resolution and irregular, PE-Spatial features are moderately noisy but informative, and DINOv2/DINOv3 features are spatially consistent and well structured. Additional visualizations are shown in Figure 15.

Figure 4 visualises backbones feature maps with PCA. CLIP features show low spatial detail and irregular patterns, while PE-Spatial features are noisy but still discriminative. DINOv2 and DINOv3 produce consistent, high-resolution features, aligning with their stronger performance. Figure 5 illustrates the resulting predictions across backbones, alongside memory-bank exemplars with their corresponding PCA feature visualisations. More examples for feature and model outputs comparison are provided in Figure 15 and Figure 16 respectively.

These results demonstrate that our method is not tied to specific backbone and can operate with a range of foundation encoders, and benefit from future foundation models releases.

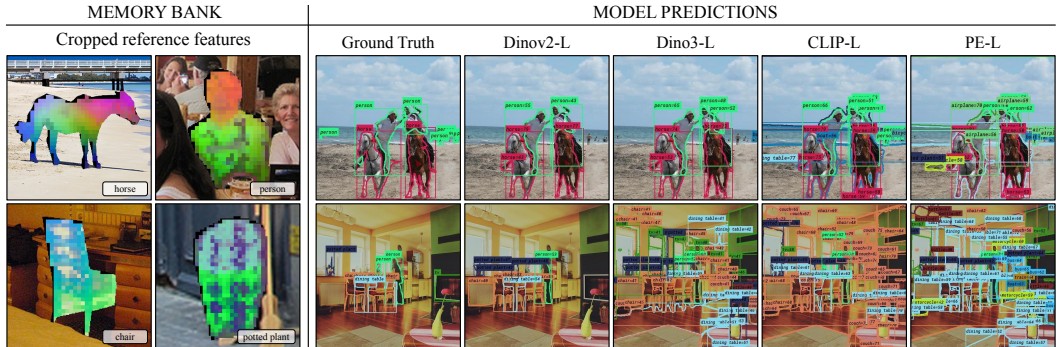

Figure 5: *Memory-bank visualisation and model predictions across backbones.* (Left) Memory-bank: reference images overlaid with their masked features. (Right) Model predictions: outputs from our pipeline with different backbones (DINOv2, DINOv3, CLIP, and PE-Spatial) along with ground truth masks for comparison. These examples illustrate how differences in feature quality influence downstream detection and segmentation. Additional outputs are provided in Figure 16.

## A.2 EVALUATE VLM MODEL ON COCO FEW SHOT DATASET

To compare with recent VLM models, we implement a joint pipeline that combines Qwen2.5-VL-7B Wang et al. (2024) and SAM2. We run Qwen2.5-VL-7B on COCO validation set to produce category-conditioned bounding boxes, which are then directly used to prompt SAM2 to obtain segmentation masks. We evaluate both detection (nAP) and segmentation performance using the same COCO few-shot class split as our method. Table 9 shows the results obtained, highlighting the performance of our model.

| Backbone models | Num shots | bbox | | | segm | | |
|---|---|---|---|---|---|---|---|
| | | nAP | nAP50 | nAP75 | nAP | nAP50 | nAP75 |
| dinov2-vitl14-pretrain + sam2-hiera-l | 30 | 36.8 | 54.5 | 38.7 | 34.2 | 54.4 | 36.7 |
| Qwen2.5-VL-7B-Instruct + sam2-hiera-l | - | 6.2 | 8.4 | 6.9 | 5.9 | 8.4 | 6.8 |

Table 9: Comparison between our DINOv2+SAM2 pipeline and the Qwen2.5-VL-7B+SAM2 baseline. Qwen-VL outputs bounding boxes that are fed to SAM2 for segmentation, while our method uses DINOv2 features with the same SAM2 decoder. Our approach achieves substantially higher detection and segmentation nAP across all metrics.

## A.3 REFERENCE-IMAGE ABLATION

To understand why different reference images lead to performance variation, we analyse the statistical properties of COCO novel classes annotations, evaluate per-class reference-image sensitivity, and design simple heuristics for selecting higher-scoring references.

### A.3.1 REFERENCE MASKS ANALYSIS

We study three annotation characteristics that intuitively affect prototype quality: (1) mask area (object size), (2) mask center location, and (3) distance to image edges.

**Mask area.** Figure 6 shows the distribution of mask areas. The distribution is strongly skewed toward small masks, confirming that many annotations contain limited visual detail.

**Mask center location.** Figure 8 shows 2D heatmaps of bounding-box centers. Most classes exhibit a centered bias, while some (e.g., car, chair) are more distributed along the horizontal axis.

**Mask distance to edge.** Figure 7 shows distance-to-edge histograms. A fraction of annotations overlap with image boundaries, possibly implying that the object is partially cropped.

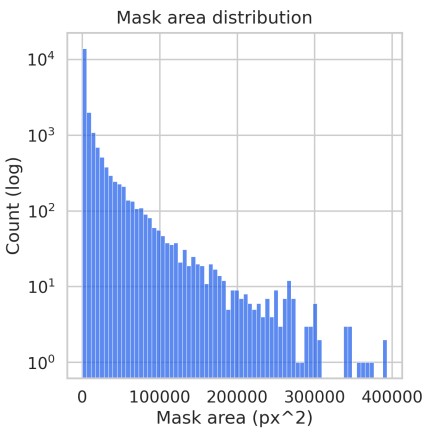

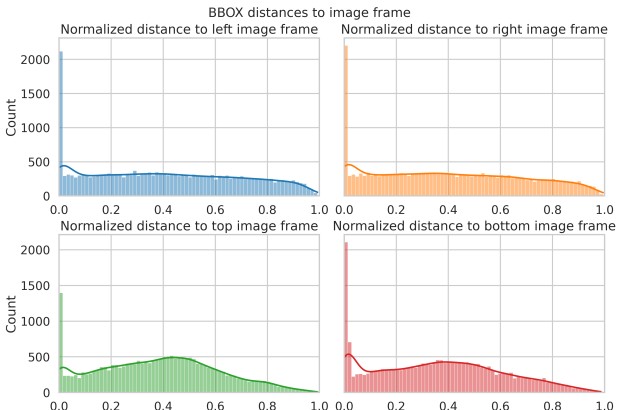

Figure 6: *Mask-area distribution for the 20 COCO novel classes*. Most annotations are small or medium-sized, as shown by the positively skewed distribution (log-scale y-axis).

Figure 7: *Distance from each mask to the four image boundaries*. A substantial number of annotations lie close to image edges, indicating possible partial cropping or truncated objects.

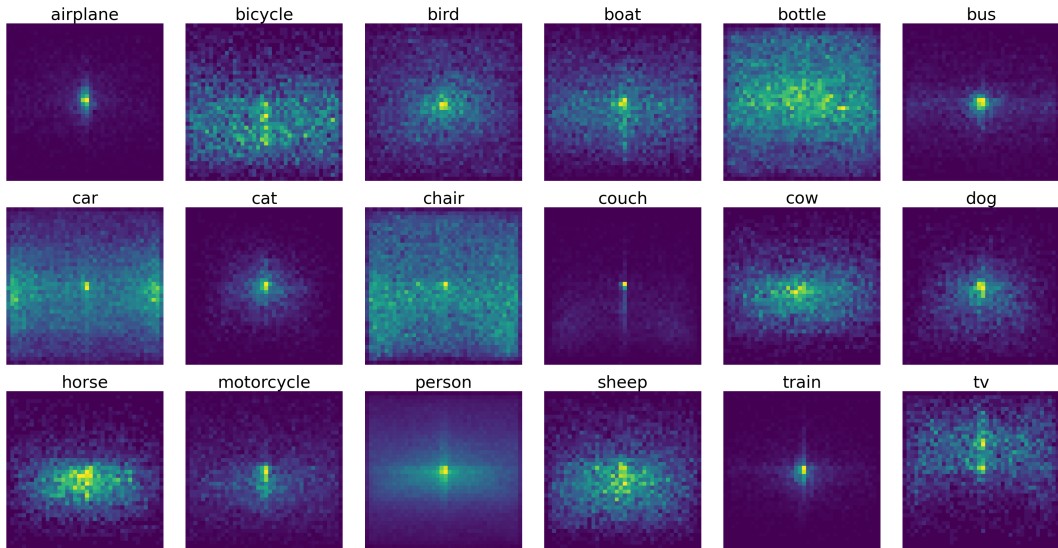

Figure 8: *2D heatmaps of mask centers across COCO novel classes*. Most annotations are near the image center, though some classes show strong horizontal spread (e.g., car, chair).

### A.3.2 HEURISTICS FOR REFERENCE SELECTION

In order to identify common patterns linked to higher performance, we sample 100 diverse reference images per class, explicitly covering a range of mask sizes, centers, and edge distances. Each reference is evaluated on a fixed reduced validation subset.

Figure 9 shows per-class scatter plots of mask area vs. performance. All classes show a clear positive trend: larger masks consistently produce stronger reference prototypes which result in higher downstream scores.

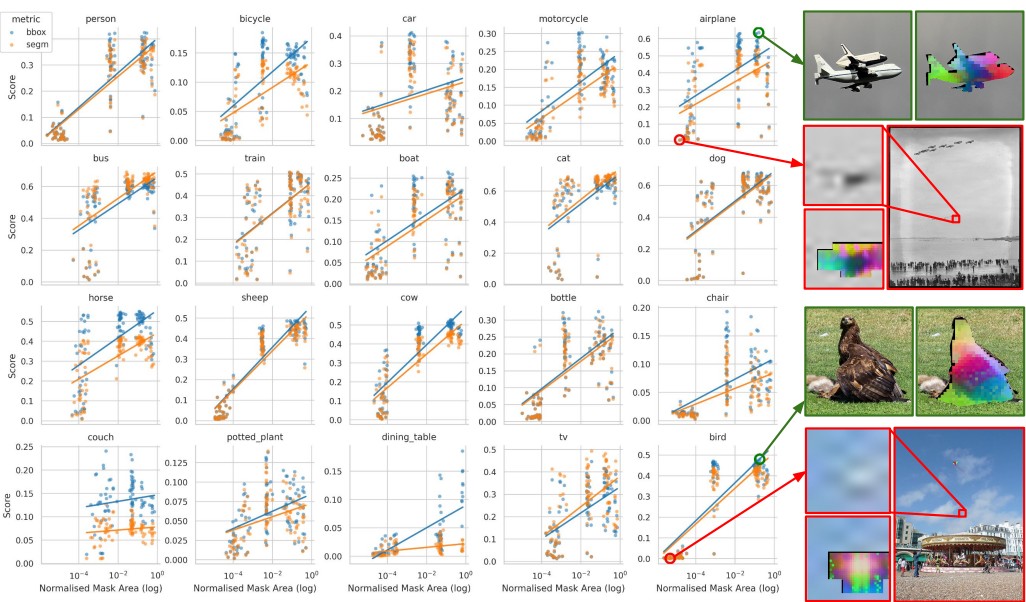

Figure 9: *Reference-image performance vs. mask area for all COCO novel classes*. Each point represents one reference image evaluated on a class-specific reduced validation subset. Larger mask areas consistently correlate with higher scores. Right: visual examples comparing a high-performing and a low-performing reference for two classes. Best viewed when zoomed in.

Based on these observations we define simple selection heuristics:

- Area category (based on class-specific quartiles): large ($\geq$75th), medium (25th–50th), small ($\leq$25th).
- Centeredness: mask center within $\pm$10% of the image center
- Edge avoidance: mask must be at least $d$ pixels away from any image boundary.

Figure 10 and Figure 11 confirm that medium and large references outperform small ones, and centered references outperform off-center ones. Distance-to-edge has weaker effect once area and centeredness are controlled.

Overall, the proposed selection heuristics effectively improve one-shot performance without increasing the number of shots, providing a practical method for choosing high-value references.

### A.3.3 REFERENCE-IMAGE DEGRADATION / METHOD ROBUSTNESS

We evaluate our method under progressively degraded reference images by applying increasing levels of Gaussian blur. For efficiency, we conducted this experiment on a small representative subset, since our goal is to measure relative performance degradation rather than absolute metrics.

As shown in Figure 12, our method remains robust even under strong blur. We attribute this stability to the invariance and consistency of DINOv2 features, which remain discriminative despite significant image degradation.

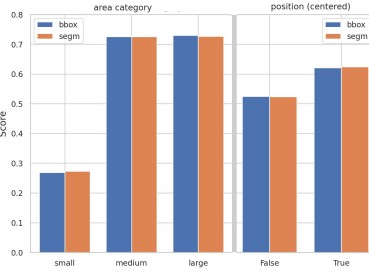
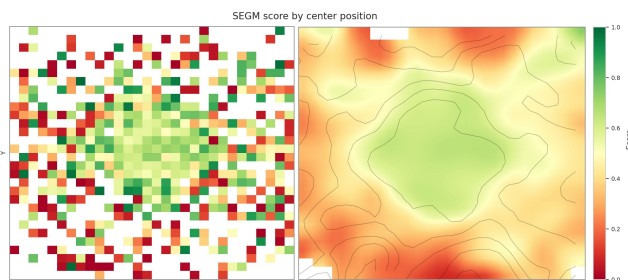

Figure 10: *Effect of mask area (left) and centeredness (right) on downstream performance.* Medium/large masks significantly outperform small masks, and centered references outperform off-center ones.

Figure 11: *2D score maps of performance as a function of mask-center location.* Both the discrete and smoothed KDE versions show higher performance when reference masks lie near the image center.

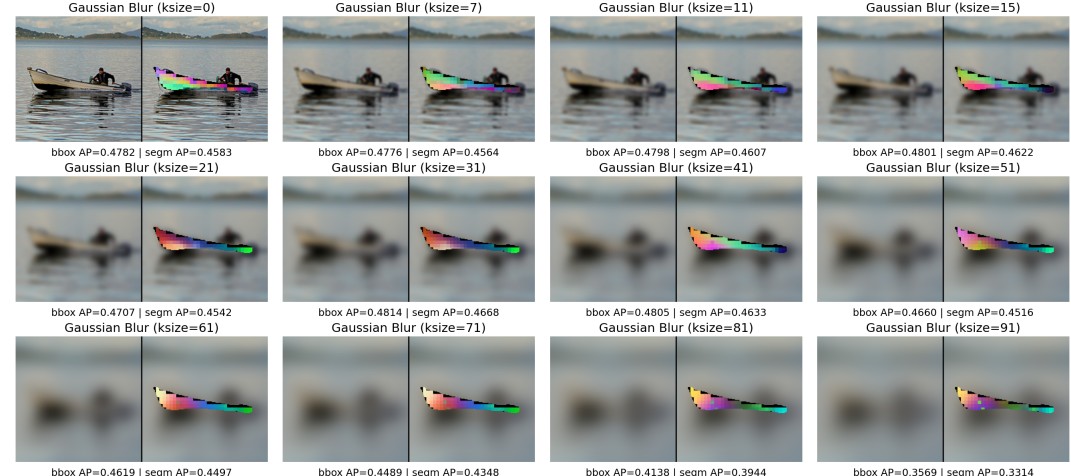

Figure 12: Model performance with increasing levels of blurring in the reference image. Each column shows a reference image blurred with an increasing Gaussian kernel size, along with its corresponding DINOv2 PCA feature visualization. Blurred input images result in blurred and smoother Dinov2 features. Below each pair, we display the predicted bounding box and segmentation mask. Our method exhibits strong robustness even under heavy blur.

## A.4 TWO-STEP AGGREGATION ABLATION

We ablate the aggregation strategy by comparing (1) class-only aggregation (direct feature averaging across all class pixels) and (2) our two-step strategy, where we first compute instance-level prototypes (pixel-weighted means within each mask) and then average them to obtain the class prototype.

As shown in Table 10, the performance difference between the two strategies is negligible. This is expected: when instances have similar pixel counts, both methods produce nearly identical prototypes. The two-step version is slightly more faithful to instance structure and therefore remains our default choice, but both strategies are equally valid.

| Backbone models | Aggregation strategy | 10-shot | | | | | | 30-shot | | | | | |
| | | bbox | | | segm | | | bbox | | | segm | | |
| | | nAP | nAP50 | nAP75 | nAP | nAP50 | nAP75 | nAP | nAP50 | nAP75 | nAP | nAP50 | nAP75 |
| dinov2-sam2 | class agg | 35.7 | 52.5 | 37.6 | 33.4 | 52.5 | 35.9 | 36.8 | 54.4 | 38.7 | 34.4 | 54.4 | 36.9 |
| dinov2-sam2 | class+inst agg | 35.7 | 52.6 | 37.6 | 33.3 | 52.5 | 35.8 | 36.8 | 54.5 | 38.7 | 34.2 | 54.4 | 36.7 |

Table 10: Ablation of two-step prototype aggregation strategy. We compare class-only aggregation (single-step) with two-step aggregation (instance-level pixel-weighted prototypes followed by class averaging). Results are reported for 10-shot and 30-shot settings using nAP, nAP50, and nAP75.

## A.5 DINOv2 Feature Analysis

We compute DINOv2 features for all annotated objects in the COCO validation set (5k images) and visualize them using t-SNE. Figure 13 shows representative class pairs and triplets with varying degrees of semantic similarity.

Easily distinguishable classes (e.g., cat–dog, airplane–bear) form well-separated clusters. In contrast, semantically similar categories (e.g., car–truck, chair–couch, wine glass–cup–vase) exhibit substantial embedding overlap. This indicates that similar-class confusion is largely driven by DINOv2's feature geometry rather than prototype construction.

Improving performance in such cases requires stronger semantic disentanglement at the backbone level, which remains an open research direction.

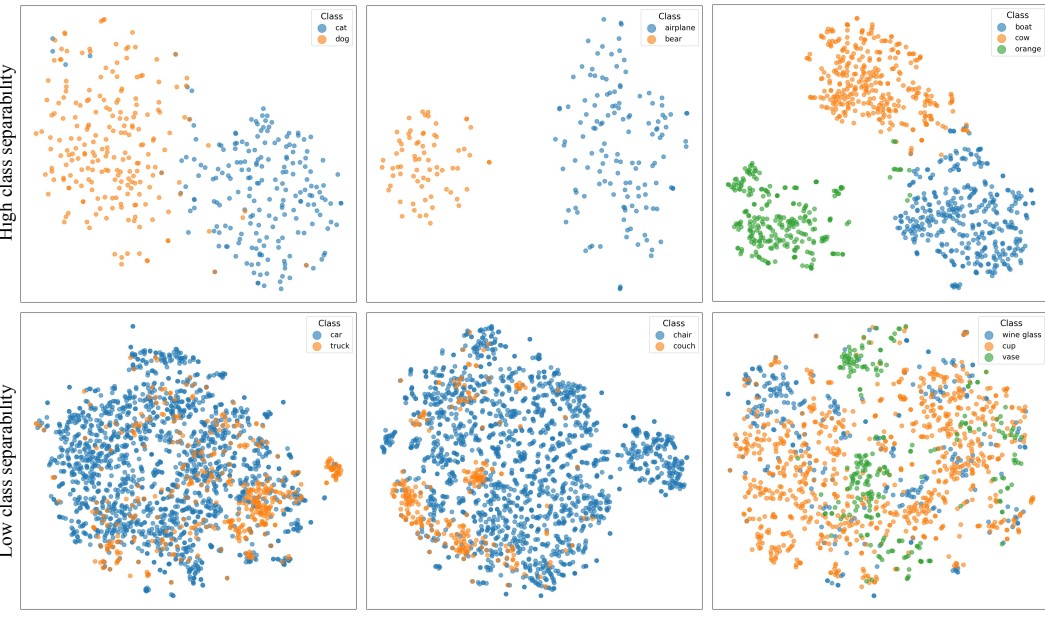

Figure 13: *t-SNE visualizations of DINOv2 object-level features for selected COCO classes.* Top row: pairs of classes with clear separability (cat–dog, airplane–bear, boat–cow–orange). Bottom row: semantically similar classes with substantial embedding overlap (car–truck, chair–couch, wine glass–cup–vase). These results indicate that similar-class confusion primarily originates from DINOv2's feature-space entanglement, limiting the effectiveness of alternative prototype-construction strategies, and setting an upper bound for our method performance.

### A.6 FEATURE SIMILARITY

To better illustrate how our feature-matching pipeline operates, we visually analyse DINOv2 feature similarity at the patch level and at the prototype level. Given a reference mask, we extract its DINOv2 features and visualize them via PCA. (see Fig. 4). We then compare two forms of similarity:

- **Single-feature similarity (patch feature)**. We select one feature vector inside the reference mask and compute *intra-class* cosine similarity within the same image, and *inter-class* similarity with a target image. These maps often highlight only the corresponding part of the object (e.g., the neck of a dog), showing that single features capture local appearance but lack full object coverage.

- **Prototype similarity (aggregated features)**. We average all features inside the reference mask to obtain a class-level prototype. When computing intra-class and inter-class similarity with this prototype, similarity maps become more spatially coherent and object-aligned. This confirms that averaging features over the entire object yields a more stable representation for matching instances in the memory bank.

Figure 14 shows one example comparing both the single-feature and aggregated-feature similarity. Figures 17 and 18 provide additional paired examples across more images and classes.

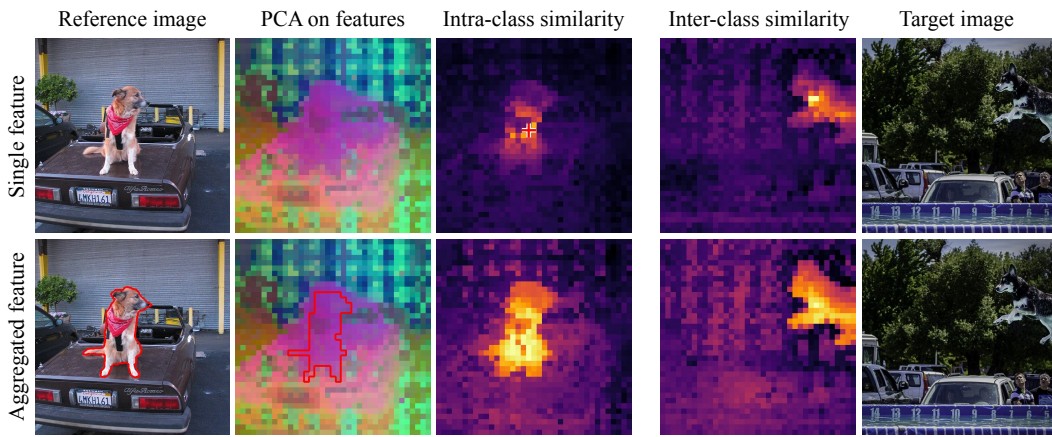

Figure 14: *Single vs. aggregated feature similarity.* We compare cosine similarity maps obtained from (top) a single DINOv2 patch feature selected inside the reference mask (marked with **+**) and (bottom) the aggregated prototype obtained by averaging all features within the mask. For each case, we show intra-class similarity (within the same image) and inter-class similarity (with a target image). Single-feature similarity highlights only local object parts, whereas aggregated features produce more coherent, object-level similarity patterns.

### A.7 ADDITIONAL FIGURES

#### A.7.1 FEATURE COMPARISON ACROSS SEMANTIC BACKBONES

In Figure 15, we provide extended examples comparing the PCA projections of DINOv2, DINOv3, CLIP, and PE-Spatial features, showing spatial structure and noise characteristics across backbones.

Figure 16 includes extended model output visualizations for multiple semantic backbones, highlighting how differences in feature quality propagate to detection and segmentation predictions.

Figures 17 and 18 contain examples of similarity maps for both single-feature and aggregated-feature settings across more images and classes.

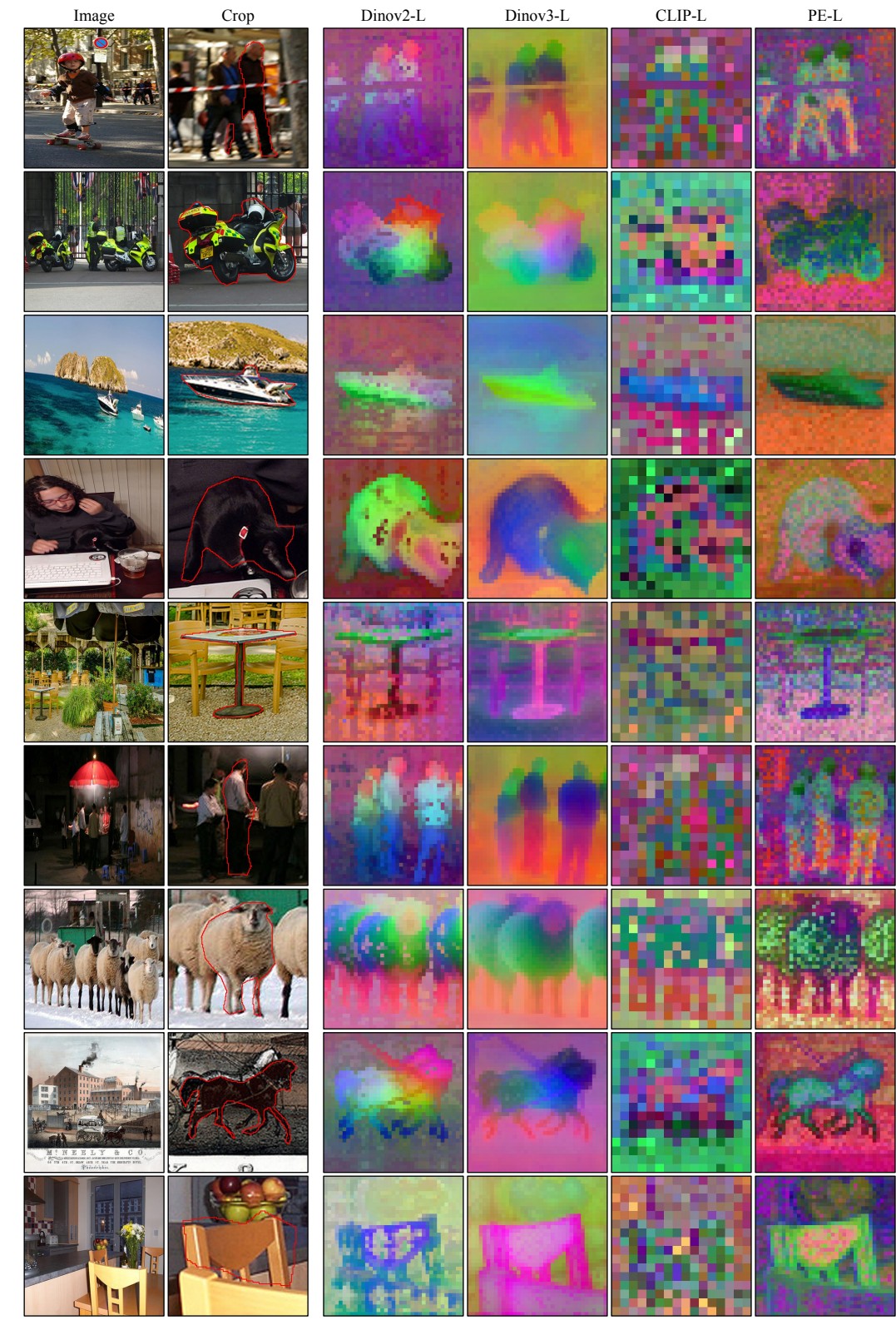

Figure 15: *Extended feature comparison across semantic backbones.* PCA projections of DINOv2, DINOv3, CLIP, and PE-Spatial feature maps for additional examples. These visuals highlight qualitative differences in spatial resolution, noise, and feature consistency.

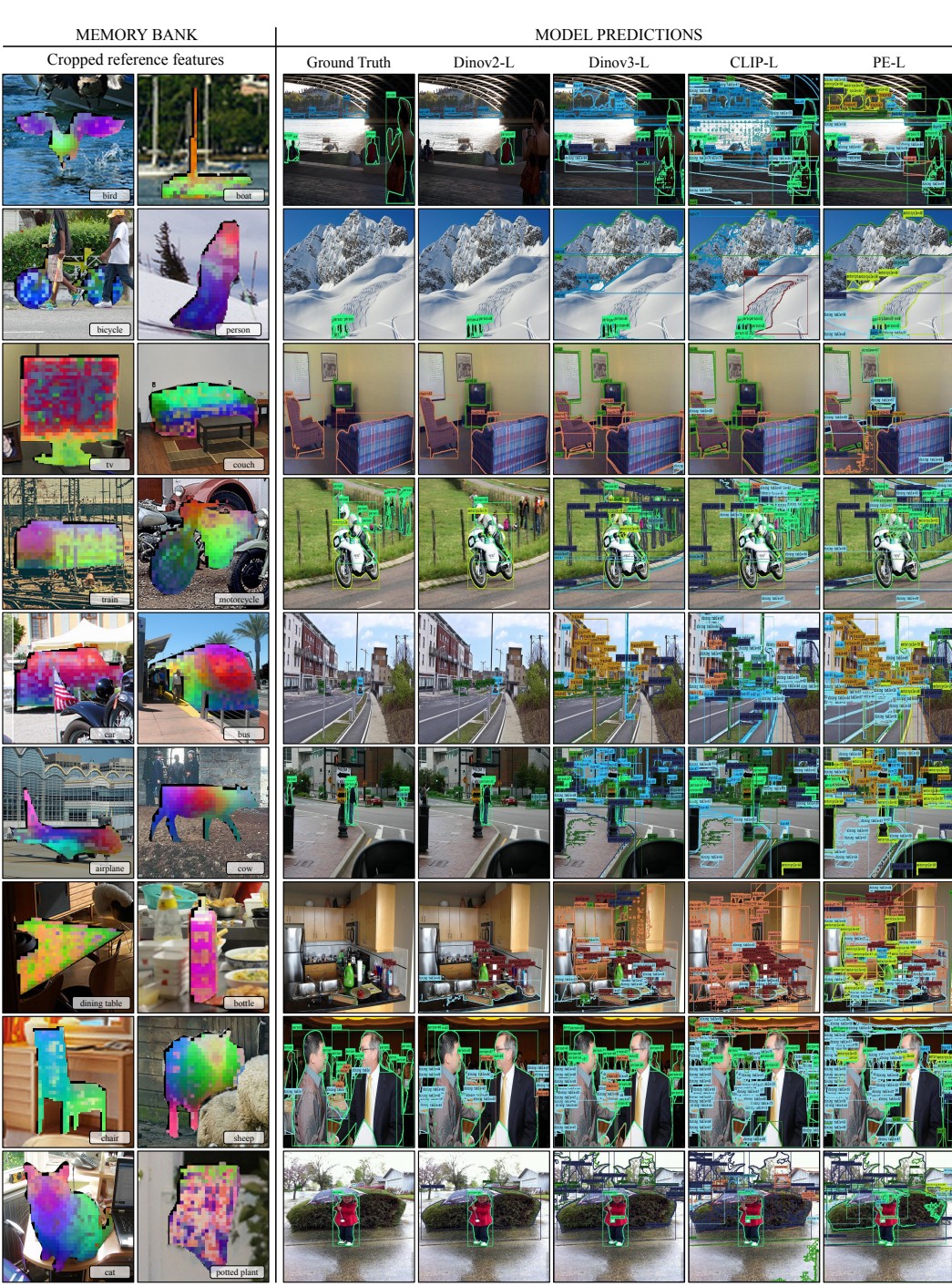

Figure 16: *Extended output comparison across semantic backbones.* Predictions from multiple backbones on additional images, including ground-truth masks and corresponding model outputs. Differences in feature quality lead to distinct segmentation and detection behaviours. Best viewed when zoomed in.

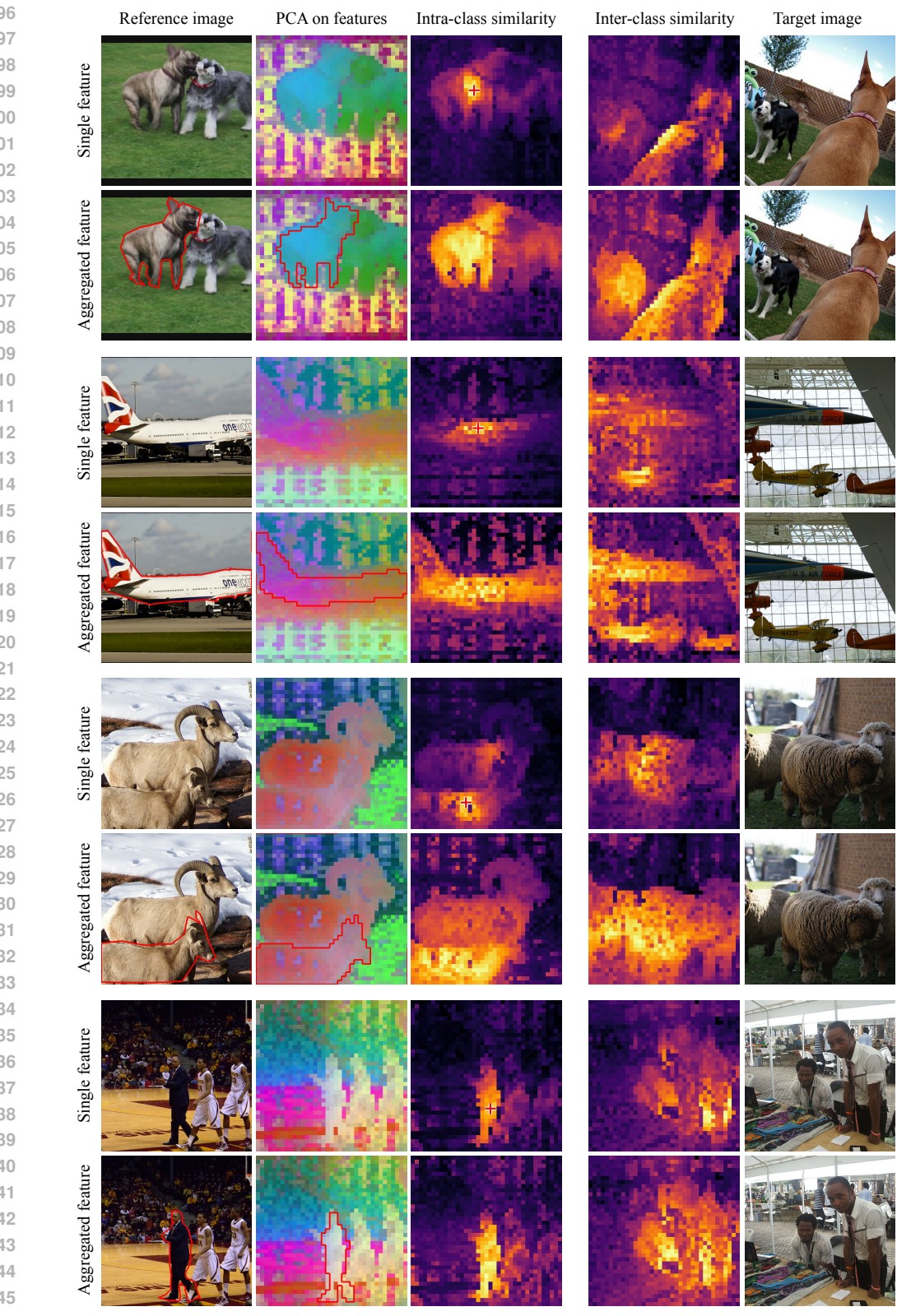

Figure 17: *Feature similarity — additional paired examples.* For each reference-target image pair: the top row shows cosine similarity maps computed from a single DINOv2 patch feature (marked +), and the bottom row shows maps computed from the aggregated prototype (mask-area average). For each row we display intra-class (same image) and inter-class (target image) similarity. Paired examples demonstrate that aggregated prototypes yield more coherent, object-level similarity than single patch features.

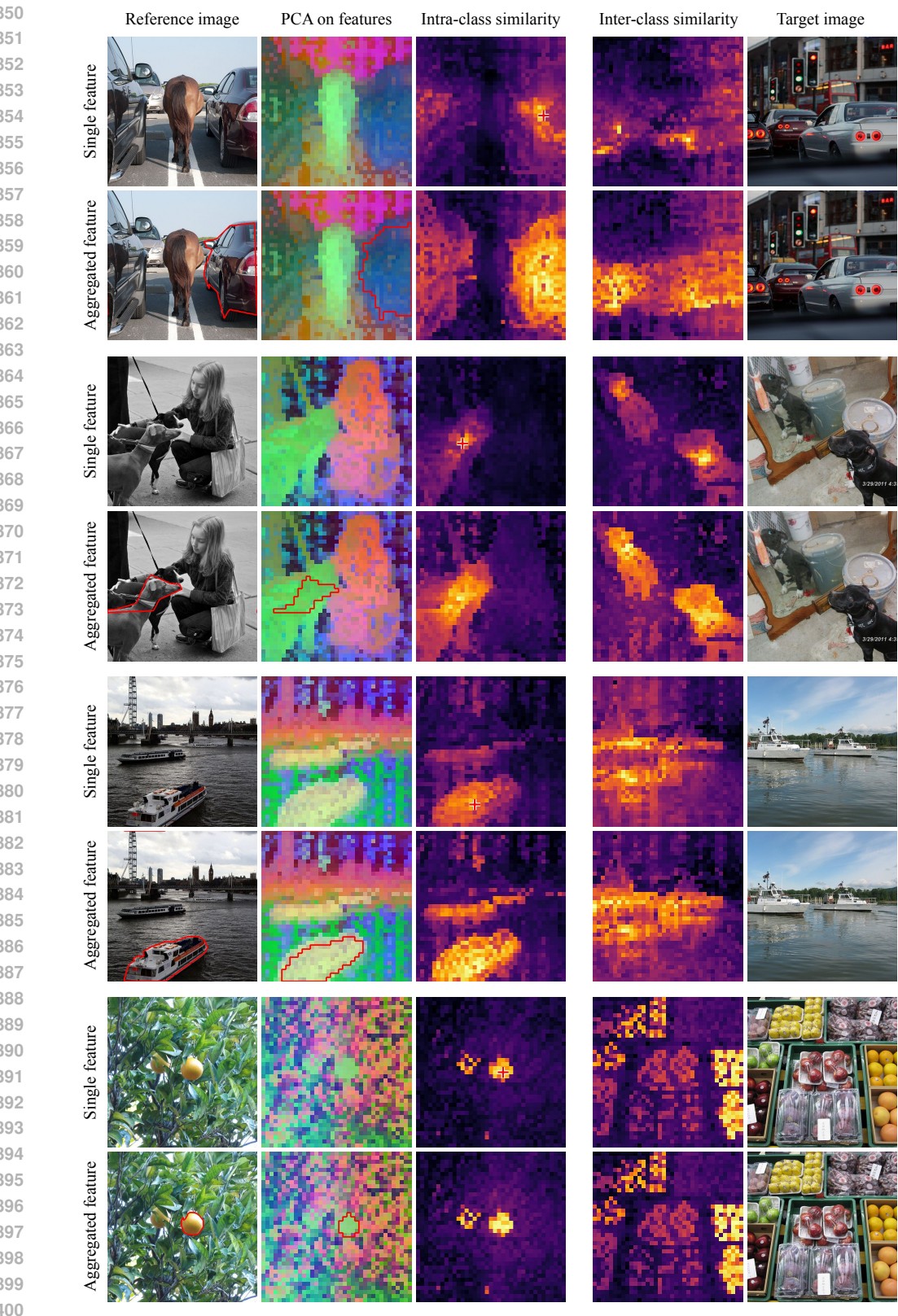

Figure 18: *Feature similarity — extended paired examples.* For each reference-target image pair: the top row is single-feature similarity (patch +) and the second row is prototype similarity (mask-average). Each pair shows intra- and inter-class cosine similarity. These examples reinforce that aggregation produces more stable and spatially consistent instance-level matches.

### A.8 DINOv3 ANALYSIS

**Feature separability.** Figure 19 shows t-SNE embeddings for DINOv2-L and DINOv3-B/L/H. Classes that are not well-separated in DINOv2 remain similarly entangled in all DINOv3 variants, indicating that the semantic structure relevant for prototype-based matching does not substantially improve in DINOv3. Figure 20 provides complementary PCA visualisations of the spatial feature maps, showing that DINOv3 produces smoother and more spatially uniform features, while DINOv2 exhibits sharper local variation. These results suggest that although DINOv3 representations are denser, their object-level semantics remain comparable to DINOv2, which may limit performance improvements in our training-free setting.

**Backbone size effects.** Figure 21 compares model outputs across backbones. DINOv3 backbones (B/L/H) tend to assign high similarity scores more broadly across proposals.

### A.9 CROSS-DOMAIN FEW-SHOT OBJECT DETECTION

We provide additional visualisations of the six target datasets in the CD-FSOD benchmark. These datasets span diverse and challenging domains, including photorealistic, cartoon, aerial, underwater, and industrial imagery, each presenting unique distribution shifts. Despite these variations, our method achieves strong performance across all domains without any fine-tuning, demonstrating its remarkable cross-domain generalization. Figure 22 shows an overview of the results for the 6 datasets. Figures 23, 24, 26, 25, 27, 28 display more detailed results for each of the datsets. All results are shown for 5-shot setting, using 5 reference images per category.

### A.10 COCO-$20^i$

Despite the strong performance of our training-free method across datasets, it also exhibits certain limitations, displayed in Figure 29. A recurring failure mode is the confusion between semantically similar categories, such as bread being misidentified as a hot dog or large armchairs being mistaken for couches. This suggests that our approach could benefit from more fine-grained differentiation or improved selection of reference images. Additionally, detecting small or fine-grained objects remains challenging, as some instances are missed. Finally, in densely crowded scenes, where multiple overlapping objects appear, our model tends to under-detect instances, likely due to occlusions and the complexity of the visual context. These observations highlight areas for future improvement in robust object detection and segmentation under few-shot constraints.

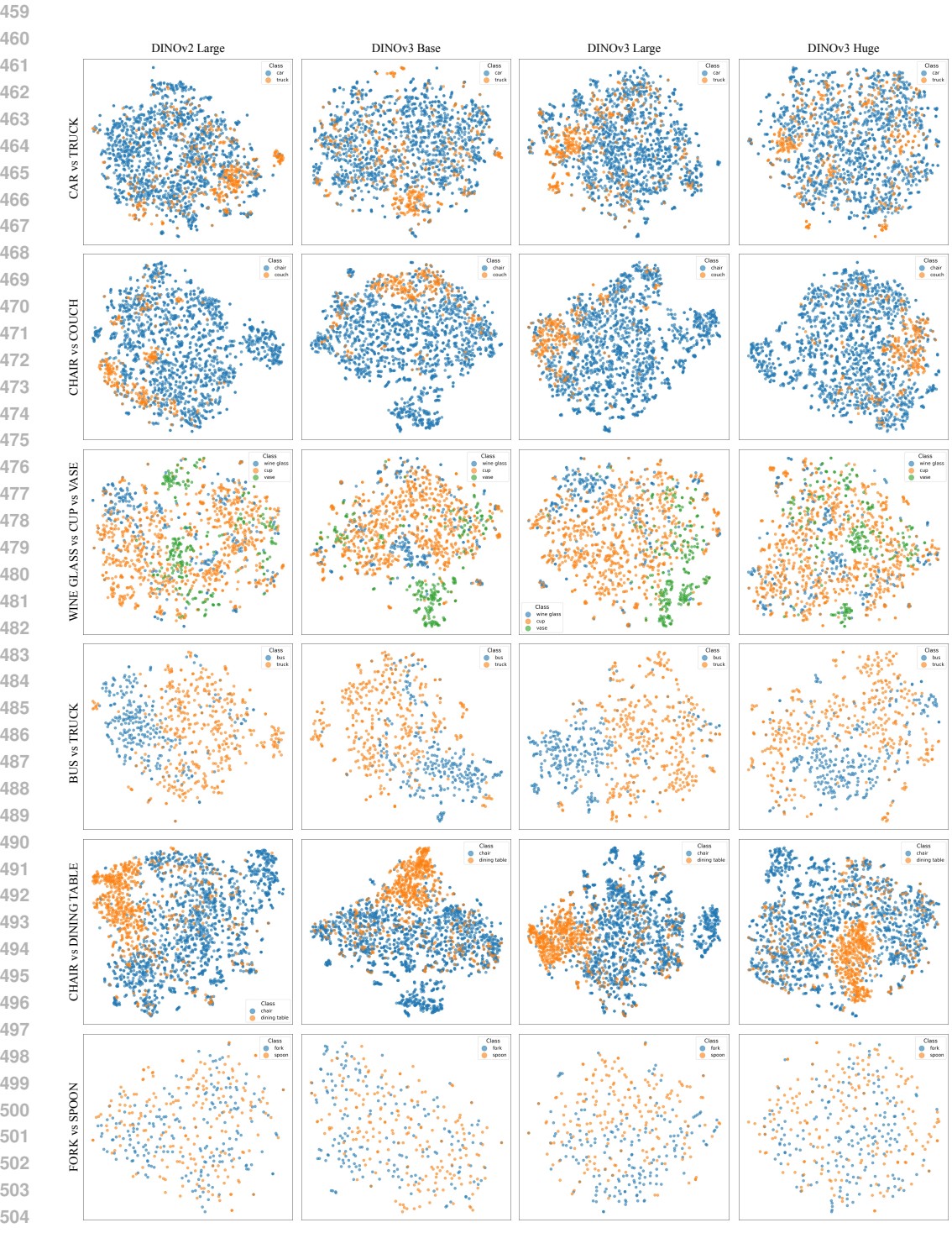

Figure 19: *t-SNE embeddings across DINO backbones.* t-SNE visualisation of feature embeddings for representative classes (e.g., fork and spoon) using DINOv2-L, DINOv3-B, DINOv3-L, and DINOv3-H. Clusters that overlap in DINOv2 remain overlapping in DINOv3 variants, indicating that semantic separability does not improve across these models.

| Image (cropped) | Dinov2-L | Dinov3-B | Dinov3-L | Dinov3-H |

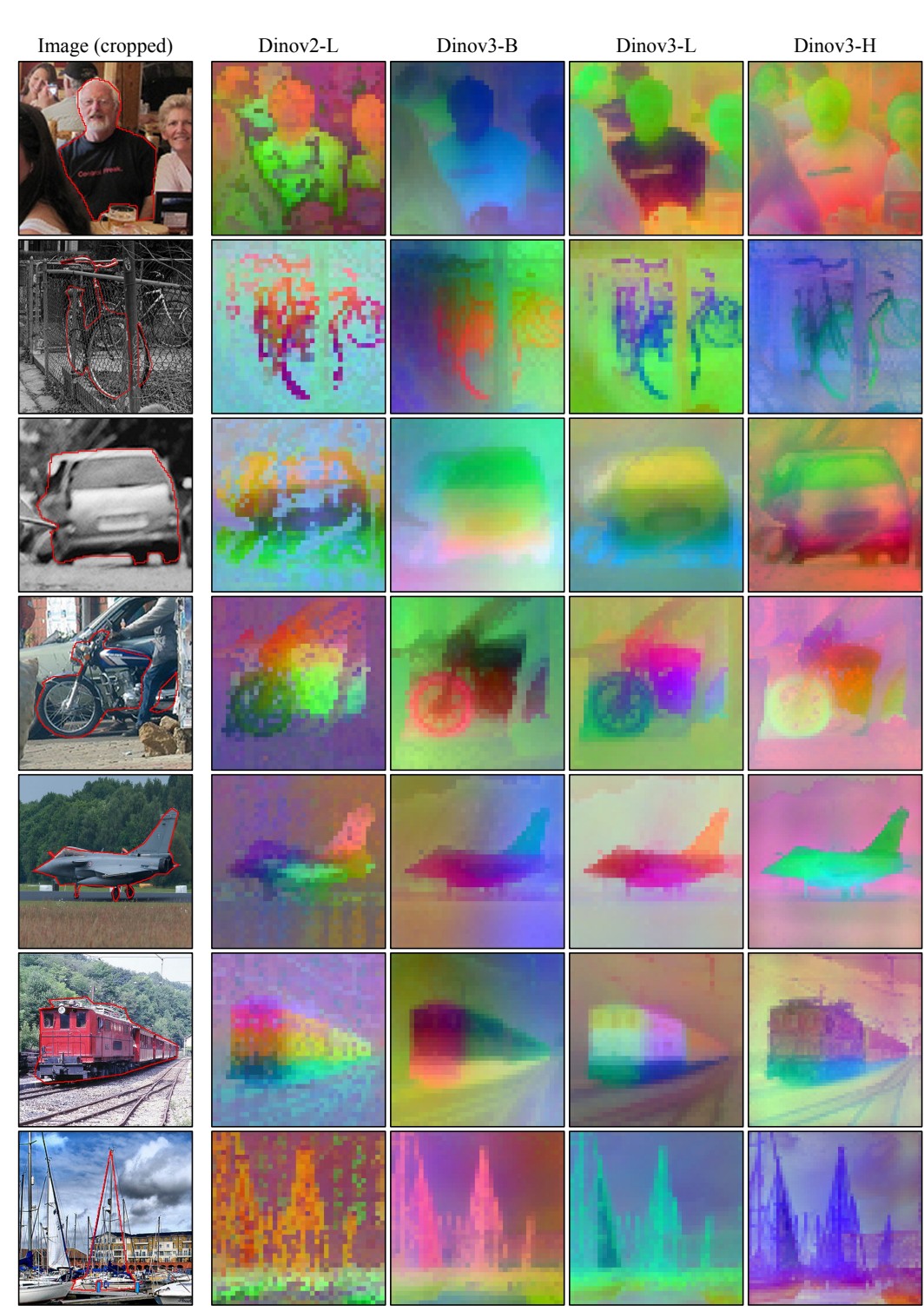

Figure 20: *Feature visualisations across DINO variants.* PCA projections of the spatial features for DINOv2-L and DINOv3-B/L/H.

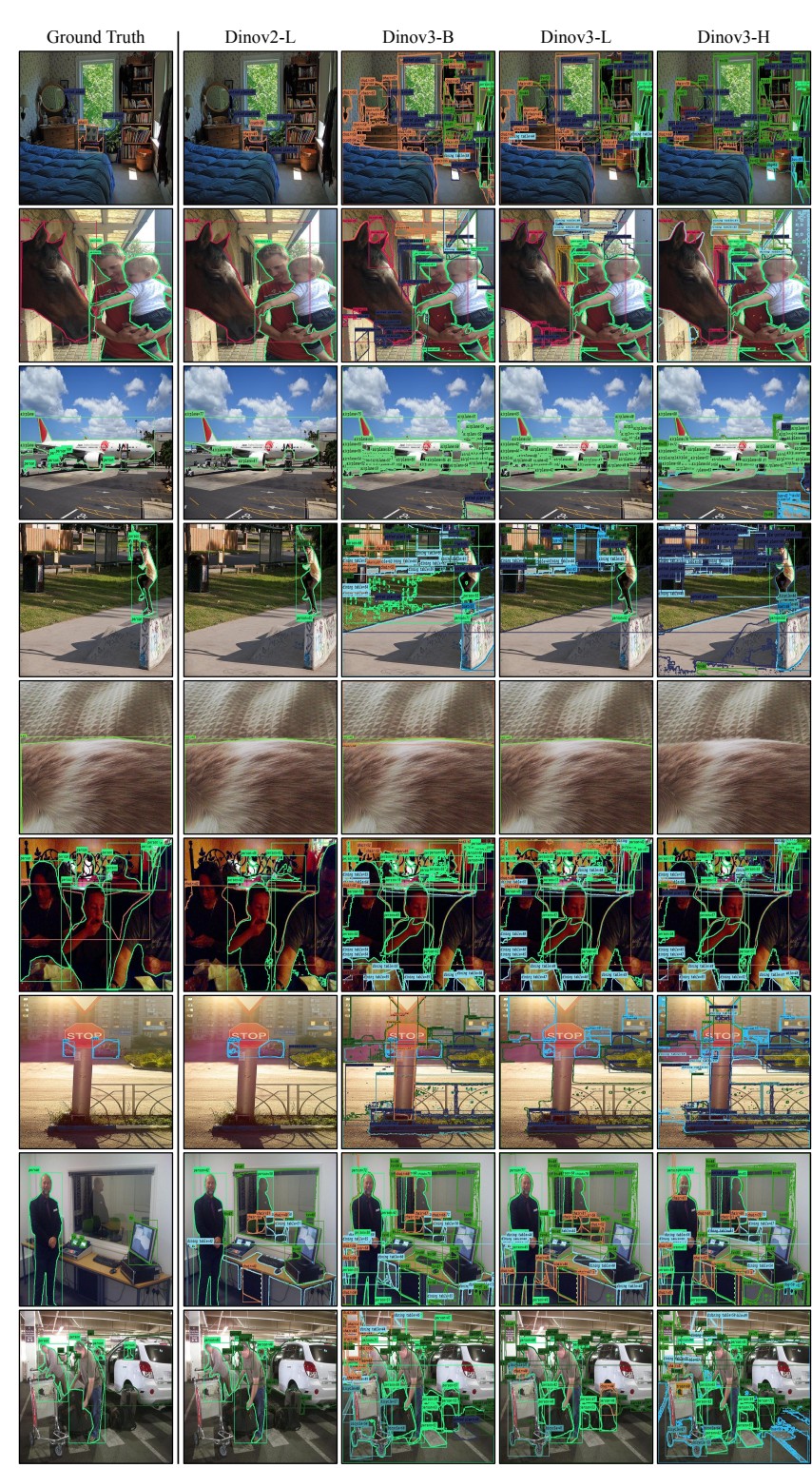

Figure 21: *Output comparison across DINOv3 backbones.* Comparison of training-free outputs for DINOv2-L and DINOv3-B/L/H. In DINOv3 models, higher similarity scores are often assigned to multiple proposals, and predictions show less object-focused contrast. Ground-truth masks are shown for reference.

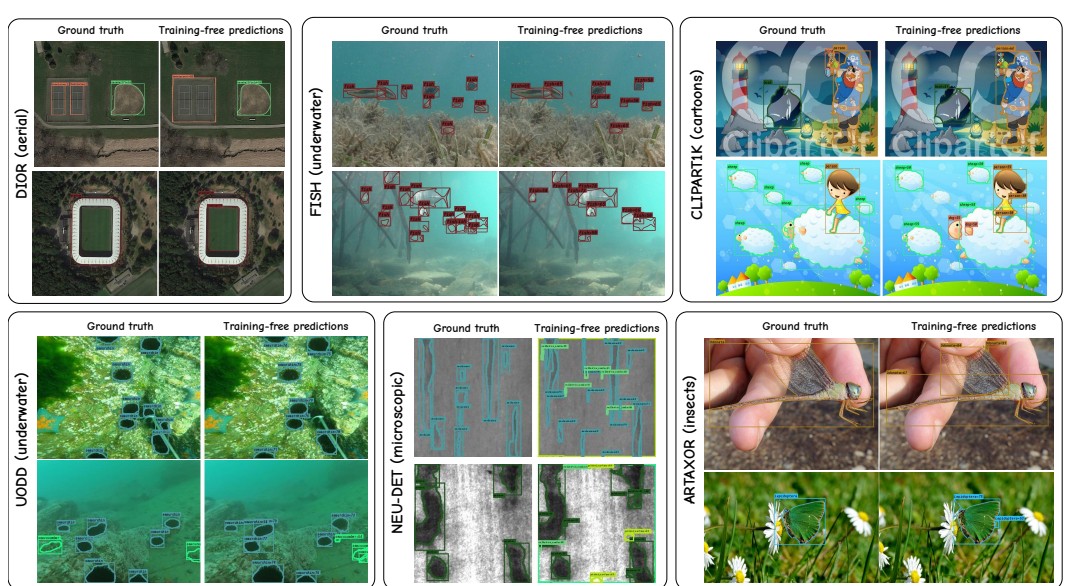

Figure 22: **Cross-domain 5-shot segmentation results using our training-free method.** Our approach evaluates diverse datasets across multiple domains, including aerial, underwater, microscopic, and cartoon imagery, without requiring fine-tuning. Results demonstrate the robustness and generalisability of our method.

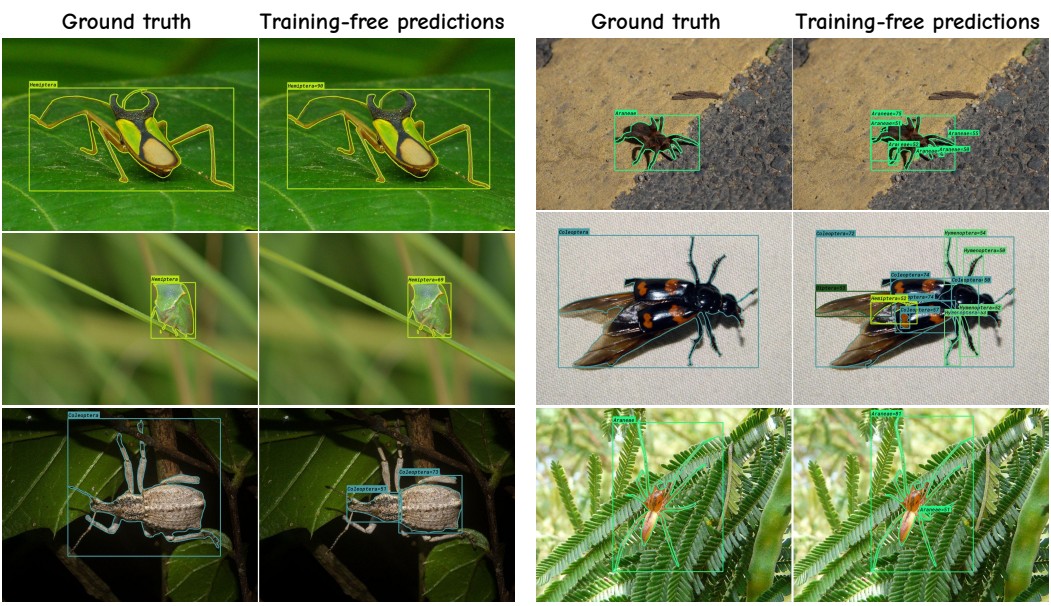

Figure 23: 5-shot results on the ArTaxOr dataset.

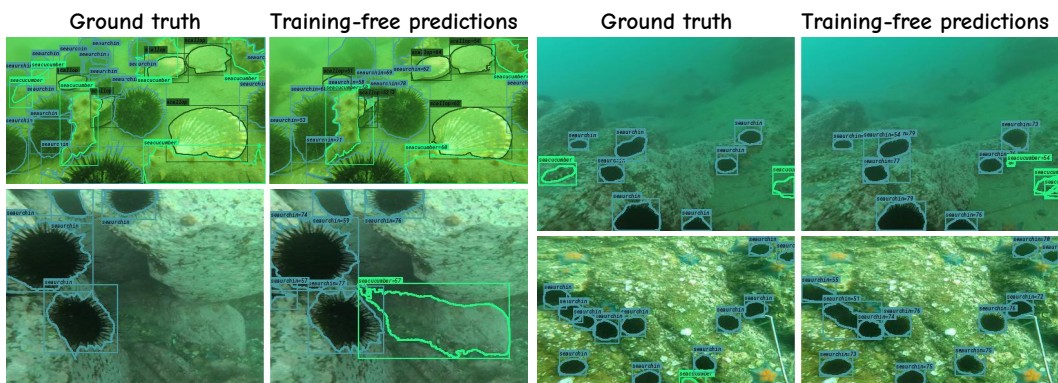

Figure 24: 5-shot results on the UODD underwater dataset.

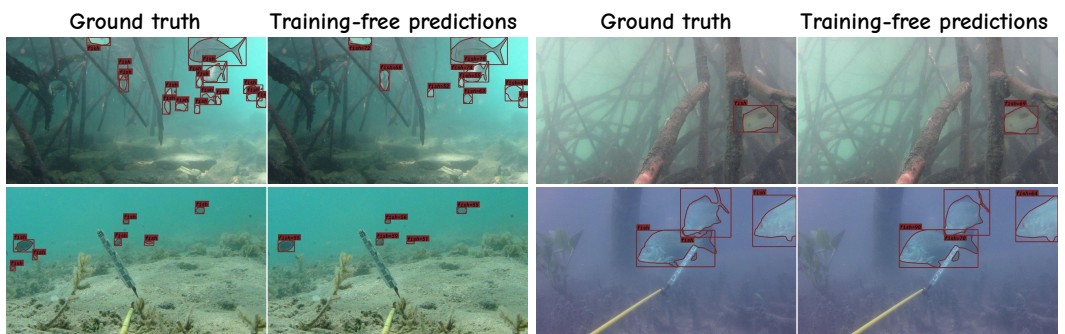

Figure 25: 5-shot results on the Fish dataset.

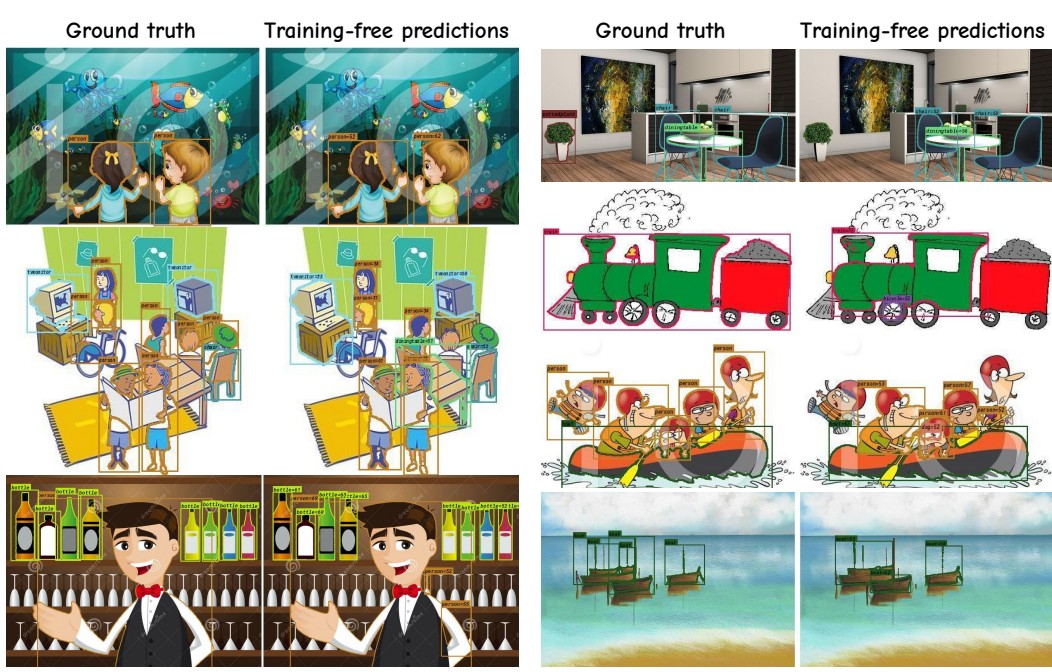

Figure 26: 5-shot results on the Clipart1k dataset.

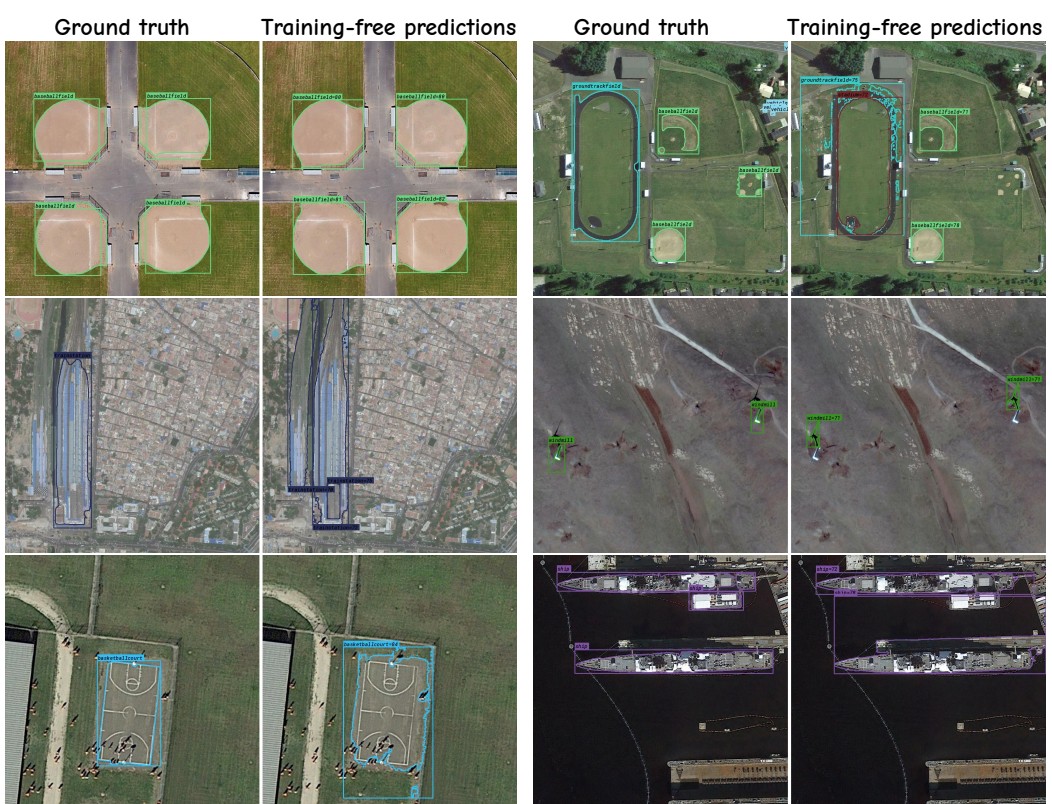

Figure 27: 5-shot results on the DIOR dataset.

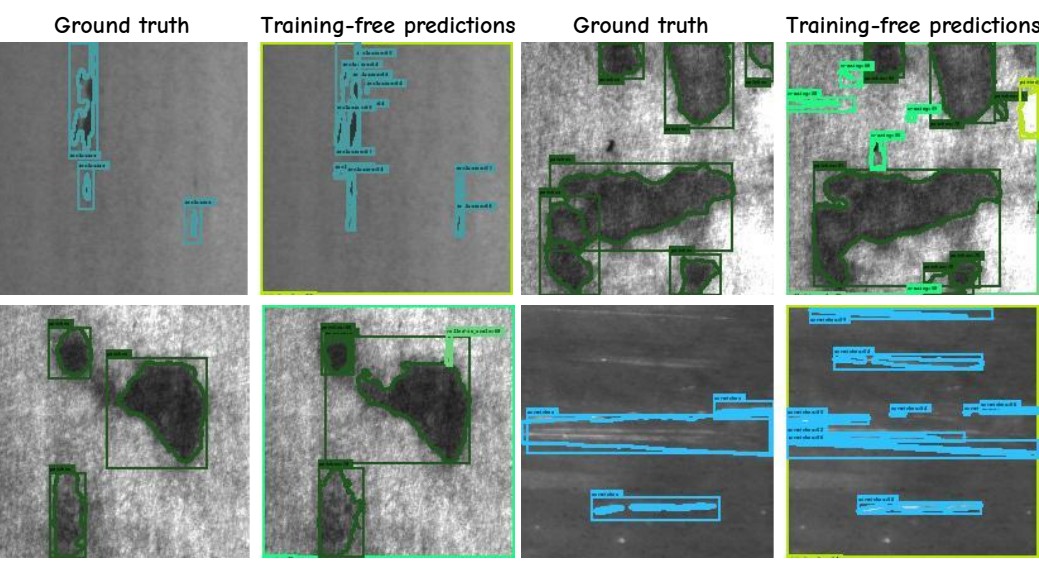

Figure 28: 5-shot results on the NEU-DET dataset.

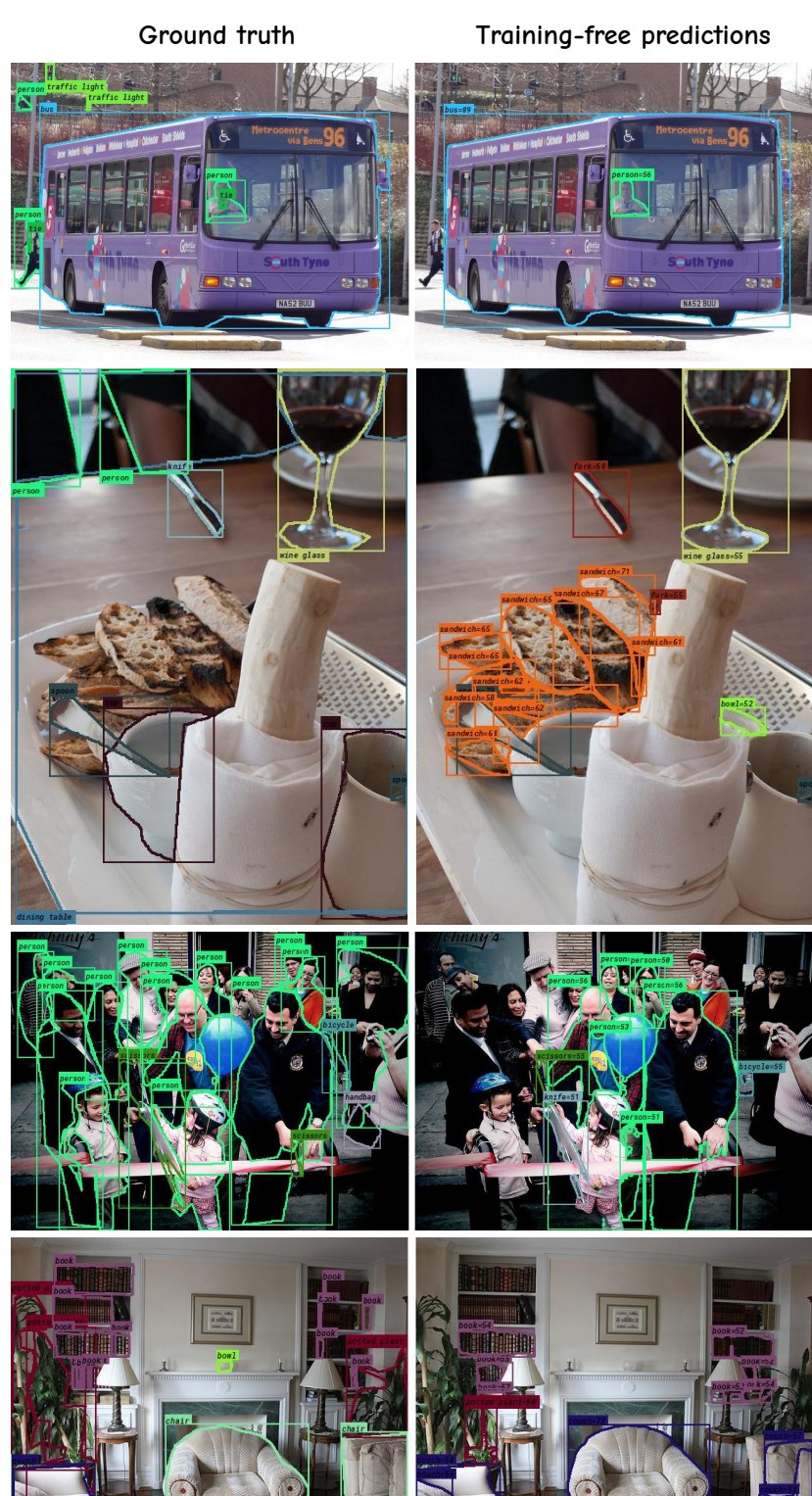

Figure 29: Visualisation of failure cases of our training-free method on the COCO val2017 set under the 10-shot setting (using 10 reference images per class). Our method sometimes confuses semantically similar classes, such as misclassifying bread as a hot dog or a large armchair as a couch. Additionally, we observe that fine or small objects are occasionally missed, and in highly crowded scenes, our model struggles to detect all instances accurately.