# OpenReview forum: "No time to train! Training-Free Reference-Based Instance Segmentation"
_ICLR.cc/2026/Conference — Submitted to ICLR 2026_

### Official Review · Reviewer_5kET · 2025-10-26

**Soundness:** 3
**Presentation:** 3
**Contribution:** 3
**Rating:** 6
**Confidence:** 4

**Summary:**

This paper proposes a training-free method for reference-based instance segmentation by leveraging two powerful pretrained vision foundation models: SAMv2 for segmentation mask proposals and DINOv2 for semantic feature extraction. The method includes a three-stage process: (1) memory bank construction from reference images; (2) feature aggregation into class prototypes; and (3) feature matching with a semantic-aware soft merging strategy for mask selection. The framework is simple, scalable, and effective, achieving state-of-the-art performance on multiple few-shot detection/segmentation benchmarks without any additional training.

**Strengths:**

1.The paper is well-written and clearly structured. The motivation is timely and compelling—reducing annotation and training cost is highly relevant in the era of foundation models.

2. The method is elegant and efficient, with minimal overhead and no need for finetuning, making it broadly applicable to low-resource or rapid-deployment scenarios.

3. The method performs competitively (or even better) than fine-tuned approaches on COCO-FSOD, PASCAL-FSOD, and CD-FSOD, with good cross-domain generalization.

4. Simple but effective use of DINOv2 + SAM: The paper demonstrates how powerful pretrained models can be combined in a modular, reproducible way to tackle challenging tasks like instance segmentation.

**Weaknesses:**

1. Limited discussion of some generalist model such as DINO-X, SINE, and T-REX:
These methods seems to also support reference-based Instance Segmentation. The current paper does not systematically analyze how its method compares in terms of design, generalization, or efficiency.

2.Evaluation scope: results emphasize benchmarks; no demonstration on real-world deployment or interactive scenarios.

**Questions:**

1.Have you tested the approach using different foundation encoders (e.g., CLIP,MAE,DINOV3 vs DINOv2)? Is the method sensitive to the embedding geometry?

2.How does the method handle reference ambiguity—for example, when multiple objects of the same class vary significantly in appearance or scale?

---

> ### Author Response · Authors · 2025-11-24
> **Author response [1/2]**
>
> We thank the reviewer for their valuable feedback, and respond to each point below.
>
> ---
> ## Limited discussion
>
> We thank the reviewer for the suggestion to expand the discussion, and we update the manuscript where appropriate.
> T-REX (Lines 133-137) relies on large-scale supervised contrastive pretraining. It is explicitly pretrained on instance segmentation tasks. In contrast, our method is fully training-free and uses frozen DINOv2 and SAM2, neither of which is pretrained for reference-based instance segmentation. We agree these models are powerful; however, their architecture and training costs differ fundamentally from our plug-and-play design.
> SINE (Lines 169-173) tackles task ambiguity by disentangling segmentation tasks. However, handling multi-instance scenarios and semantic ambiguity without training remains challenging.
> DINO-X (Citation updated in manuscript, Lines 133-135) is an object-centric model with extensive pretraining for segmentation-oriented tasks. On the other hand, our method uses DINOv2, which is not pretrained for segmentation or object delineation, and SAM2, which requires prompting. Our method reuses both in a training free manner.
>
> ---
> ## Evaluation scope (Section 4.4)
>
> We thank the reviewer for the suggestion, and we update Section 4.4. Our full pipeline occupies 1.97 GiB at runtime. When using 20 classes and 10 shots per class (200 reference images) CUDA-allocated memory reaches a peak of 11.5 GiB. Inference speed is 0.93 seconds per image (x120 times faster than Matcher). These additions clarify the deployment footprint and will support future extensions toward lightweight edge-oriented variants.
>
> ---
> ## Multiple encoders (Appendix A.1)
>
> To evaluate backbone dependence, we replace DINOv2 with CLIP, DINOv3, and PE-Spatial, exploring different model sizes. Quantitative results are shown in Table A (extended results in Appendix A.1 Table 8), and qualitative feature-map visualizations in Appendix A.1 Figures 4, 5, 15, 16.
>
> *Table A. Backbone ablation on COCO-FSOD, comparing different semantic backbones*
>
> | **Semantic backbone**        | **Segm. backb.** | **10-shot bbox nAP** | **10-shot segm nAP** | **30-shot bbox nAP** | **30-shot segm nAP** |
> |------------------------------|------------------|-----------------------|------------------------|-----------------------|------------------------|
> | DINOv2-ViT-L-14              | SAM2-L           | 35.7 | 33.3 | 36.8 | 34.2 |
> | DINOv3-ViT-B-16              | SAM2-L           | 32.8 | 30.9 | 33.5 | 31.6 |
> | DINOv3-ViT-L-16              | SAM2-L           | 33.8 | 32.2 | 34.4 | 32.8 |
> | DINOv3-ViT-H-16+             | SAM2-L           | 25.5 | 24.4 | 26.2 | 25.0 |
> | DINOv3-ViT-H-16+ (px)        | SAM2-L           | 26.8 | 25.5 | 27.6 | 26.3 |
> | CLIP-ViT-B-32                | SAM2-L           | 15.2 | 13.9 | 15.8 | 14.6 |
> | CLIP-ViT-B-16                | SAM2-L           | 19.2 | 18.1 | 19.5 | 18.3 |
> | CLIP-ViT-L-14                | SAM2-L           | 18.6 | 17.3 | 18.7 | 17.4 |
> | CLIP-ViT-L-14-336px          | SAM2-L           | 17.8 | 16.7 | 17.7 | 16.7 |
> | PE-Spatial-L-14-448 (PE)     | SAM2-L           | 25.7 | 24.1 | 26.5 | 24.7 |
> | PE-Spatial-L-14-448 (IN)     | SAM2-L           | 26.5 | 24.7 | 27.3 | 25.4 |
> | PE-Spatial-G-14-448 (PE)     | SAM2-L           | 24.7 | 23.1 | 24.9 | 23.3 |
>
> Overall, our pipeline transfers across backbones with moderate performance variation. CLIP models exhibit a significant performance drop, primarily due to their low-resolution and noisier feature maps. In contrast, PE-Spatial and DINOv3 achieve competitive results with only modest degradation. Since all models were used off-the-shelf without hyperparameter adjustments, we expect these backbones to benefit from more careful integration.
>
> Figure 4 visualises backbones feature maps with PCA. CLIP features show low spatial detail and irregular patterns, while PE-Spatial features are noisy but still discriminative. DINOv2 and DINOv3 produce consistent, high-resolution features, aligning with their stronger performance. Figure 5 illustrates the resulting predictions across backbones, alongside memory-bank exemplars with their corresponding PCA feature visualisations. More examples for feature and model outputs comparison are provided in Figure 15 and Figure 16 respectively.
>
> These results demonstrate that our method is not tied to a specific backbone and can operate with a range of foundation encoders, and benefit from future foundation models releases.
>
> See Appendix A.1, Table 8 and Figures 4, 5, 15, 16 for full quantitative and qualitative results.

---

> > ### Comment · Reviewer_5kET · 2025-11-25
> >
> > Thank you for providing the additional experiments in the appendix. I am pleased to see the broader comparison across different semantic backbones, which significantly strengthens the empirical evaluation of the method. In particular, the results in **Table A** are quite intriguing and raise some questions that merit deeper discussion.
> >
> > First, it is interesting to observe that **DINOv3 consistently underperforms DINOv2** in this setting. Intuitively, one would expect DINOv3 to offer stronger representations, yet the few-shot detection and segmentation results suggest the opposite. It would be very helpful if the authors could further analyze potential reasons for this behavior.
> >
> > Second, the observation that **larger backbones (e.g., H or H+) perform worse than the L variants** is unexpected. Typically, larger models provide more expressive and higher-quality representations.
> > To help readers better understand these phenomena, I strongly encourage the authors to include **more in-depth analyses**, such as:
> > - Visualizations of feature distributions across different backbones,
> > - Qualitative comparisons of predictions produced by each backbone,
> >
> >
> > These additions would greatly enhance the interpretability of the backbone ablation and provide valuable insights into why certain backbones perform better than others in this framework. Overall, the extended experiments are appreciated, and further analysis along these lines would make the paper substantially stronger.

---

> > > ### Author Response · Authors · 2025-11-25
> > >
> > > We thank the reviewer for the valuable suggestions. We have added a dedicated section in the revised **Appendix A.8** to investigate the DINOv3 backbone behaviour, with three additional visualisations (Figure 19, 20, 21). We summarise the key additions below.
> > >
> > > ---
> > > ## Feature separability across DINOv2 and DINOv3 (Figure 19, Figure 20)
> > >
> > > We visualise t-SNE embeddings (**Figure 19**), showing that **class separability remains largely unchanged** across DINOv2 and DINOv3 variants, suggesting that the semantic structure most relevant to prototype matching does not improve with DINOv3. PCA feature-map visualisations (**Figure 20**), show that DINOv3 features are smoother and more spatially uniform, while DINOv2 exhibits sharper local variation.
> > >
> > > These results suggest that although DINOv3 representations are denser, their **object-level semantics remain comparable to DINOv2** (see t-SNE comparison, Figure 19), which may limit performance improvements.
> > >
> > > Additional feature visualisations can be seen in Figure 15.
> > >
> > > ---
> > > ## Backbone size effects (Figure 21)
> > >
> > > **Figure 21** compares outputs across backbones. We find that larger DINOv3 variants (B/L/H) tend to assign high similarity scores more broadly across proposals, which reduces contrast between candidate masks. To probe this behaviour, we apply a sigmoid to the similarity scores, suppressing mid-range responses while preserving confident matches. This small adjustment slightly improves DINOv3 performance (+0.5mAP), indicating that the **backbone can benefit from backbone-specific tuning** and that the observed gap is not inherent to the method.
> > >
> > > Additional model output visualisations can be seen in Figure 16.
> > >
> > > ---
> > > ## Feature similarity analysis
> > >
> > > Although not explicitly requested, we also point the reviewer to our feature-similarity analysis in Appendix A.6. The patch-level vs. prototype-level similarity maps (Figures 14, 17, 18) show how prototype aggregation yields more object-aligned responses. This may **help interpret some of the backbone behaviours** discussed above.
> > >
> > > ---
> > > We hope these clarifications and new visualisations address the reviewer’s questions.

---

> ### Author Response · Authors · 2025-11-24
> **Author response [2/2]**
>
> ## Reference ambiguity
>
> We agree that performance depends on the quality of the reference images and that using “high-value” references is desirable. To address this, we conduct an analysis of reference-mask properties and derive simple, effective heuristics for selecting good references (Appendix A.3).
>
> *Annotation analysis*
>
> - Mask area: distribution is heavily skewed to small objects (Figure 6).
> - Mask center location: 2D heatmaps show a strong center bias for most classes (Figure 8).
> - Distance to image edges: many masks are close to or touching image boundaries (Figure 7).
>
> *Per-class reference sensitivity.*
>
> For each novel class, we (1) sample 100 diverse reference images spanning a wide range of mask sizes, locations and edge distances, (2) evaluate each reference on a reduced validation subset. Figure 9 shows that larger mask area correlates strongly and consistently with higher nAP across all classes.
>
> *Heuristics for selecting high-value references.*
>
> Based on these observations, we propose automatic reference-selection rules.
>
> - Area: categorise masks by class-specific quartiles (small ≤25th, medium 25–50th, large ≥75th percentile), and prefer medium/large.
> - Centeredness: prefer masks whose centers lie within ±10% of the image center.
> - Edge avoidance: require a minimum distance ddd from all image borders.
>
> Figure 10 and Figure 11 show that medium and large masks significantly outperform small masks, and centered references are preferred over off-center ones. Furthermore, we find that once mask area and “centeredness” are controlled, the distance to frame has a weaker effect. These selection heuristics effectively improve one-shot performance without the need for increasing the number of shots, thus providing a practical and efficient method for choosing reference images. This evidence supports the validity of our approach in reducing performance variance while maintaining high-quality segmentation with a minimal reference set.
>
> (Appendix A.3.3 provides further robustness results under degraded/blurred references. As shown in Figure 12, we observe that DINOv2 features remain discriminative and our method is stable even under strong blur.)
>
> ---
> We hope these clarifications and new experiments address the reviewer’s concerns.

---

### Official Review · Reviewer_hGWi · 2025-10-31

**Soundness:** 3
**Presentation:** 3
**Contribution:** 2
**Rating:** 4
**Confidence:** 5

**Summary:**

Image segmentation faces high annotated data costs; SAM eases this via promptable segmentation but needs manual prompts/domain-specific rules, while existing reference-based methods require fine-tuning or costly metrics. The paper proposes a training-free three-stage framework fusing SAM (mask generation) and DINOv2 (semantics): building a reference feature memory bank, two-step feature aggregation, and cosine similarity matching with semantic-aware soft merging. It achieves SOTA on COCO-FSOD (36.8% nAP), PASCAL VOC Few-Shot (71.2% nAP50), and outperforms training-free methods on CD-FSOD (22.4% nAP).

**Strengths:**

1. Practical Significance: The training-free paradigm reduces deployment costs for low-annotation domains (e.g., underwater, microscopic imaging), aligning with real-world needs for fast adaptation.
2. Efficiency-Accuracy Balance: It outperforms prior methods (e.g., Matcher) on accuracy and runs ~129x faster (0.929s/img vs. 120.014s/img), striking a rare balance.
3. Clarity & Reproducibility: The three-stage framework is visualized clearly (Fig.2), with detailed implementation details (resolution, IoU threshold) and standard experiments enabling reproducibility.
4. Cross-Domain Robustness: It works across diverse domains (cartoon, aerial) with fixed hyperparameters, expanding application scope.

**Weaknesses:**

1. Limited Originality: The method combines off-the-shelf models (SAM/DINOv2) with classic techniques (cosine similarity, soft merging)—no novel methodology or theoretical insights, making it an engineering implementation rather than an innovation.
2. Incomplete Experiments: No ablation for two-step aggregation (e.g., instance-only vs. class-only prototypes) or memory bank design; no comparison with mainstream “VLM+SAM” pipelines, weakening competitiveness arguments.
3. Shallow Analysis: Failure cases (similar-class confusion) lack root-cause attribution (e.g., DINOv2’s bias); visualizations (Fig.3/4) only show predictions, no core process (feature similarity) visuals.
4. Weak Prior Work Context: Only compares performance with prior methods (e.g., Matcher) but no deep logic contrast (e.g., why cosine similarity outperforms Earth Mover’s Distance)

**Questions:**

1. Could you add ablation for two-step aggregation (instance-only/class-only vs. two-step) and memory bank design (e.g., CLIP vs. DINOv2) to verify module necessity?
2. Why not compare with “VLM (Qwen-VL)+SAM” pipelines? Please supplement CD-FSOD/COCO-FSOD experiments to show your method’s advantages.
3. For similar-class confusion, could you visualize DINOv2’s feature embeddings (t-SNE) to check semantic overlap? Can adjusting prototype construction mitigate this?
3. How does the method perform with low-quality reference images (blurred/occluded)? Please provide model size and GPU memory for edge deployment

---

> ### Author Response · Authors · 2025-11-24
> **Author response [1/3]**
>
> We thank the reviewer for their valuable feedback, and respond to each point below.
>
> ---
> ## Novelty
>
> We appreciate the reviewer’s concern and agree that our method builds upon existing foundation models. However, we respectfully argue that effective reuse of off-the-shelf components in a new problem setting can itself constitute a meaningful contribution, especially when the integration requires particular design choices. While prior attempts either require heavy pretraining (e.g., T-Rex, SINE), or rely on slow and complex matching pipelines (e.g., Matcher), our work shows that carefully engineered mechanisms—memory-bank construction, two-step aggregation, and semantic-aware soft merging—yield significant downstream impact. We therefore view our framework as a practical, scalable, and strong training-free alternative.
>
> This perspective is clarified in the updated manuscript, lines 102-105.
>
> ---
> ## Two-step vs class-only aggregation (Appendix A.4)
>
> We ablate the aggregation strategy by comparing (1) class-only aggregation (direct feature averaging across all class pixels) and (2) our two-step strategy, where we first compute instance-level prototypes (pixel-weighted means within each mask) and then average them to obtain the class prototype.
>
> As shown in Table A (extended results in Appendix A.4 Table 10), the performance difference between the two strategies is negligible. This is expected: when instances have similar pixel counts, both methods produce nearly identical prototypes. The two-step version is slightly more faithful to instance structure and therefore remains our default choice, but both strategies are equally valid.
>
> *Table A. Ablation of two-step prototype aggregation strategy.*
>
> | **Backbone models** | **Aggregation strategy** | **10-shot bbox nAP** | **10-shot segm nAP** | **30-shot bbox nAP** | **30-shot segm nAP** |
> |---------------------|--------------------------|-----------------------|------------------------|-----------------------|------------------------|
> | dinov2-sam2         | class agg                | 35.7 | 33.4 | 36.8 | 34.4 |
> | dinov2-sam2         | class+inst agg           | 35.7 | 33.3 | 36.8 | 34.2 |

---

> ### Author Response · Authors · 2025-11-24
> **Author response [2/3]**
>
> ## Memory bank design (Appendix A.1):
>
> To evaluate backbone dependence, we replace DINOv2 with CLIP, DINOv3, and PE-Spatial, exploring different model sizes. Quantitative results are shown in Table B (extended results in Appendix A.1 Table 8), and qualitative feature-map visualizations in Appendix A.1 Figures 4, 5, 15, 16.
>
> *Table B. Backbone ablation on COCO-FSOD, comparing different semantic backbones*
>
> | **Semantic backbone**        | **Segm. backb.** | **10-shot bbox nAP** | **10-shot segm nAP** | **30-shot bbox nAP** | **30-shot segm nAP** |
> |------------------------------|------------------|-----------------------|------------------------|-----------------------|------------------------|
> | DINOv2-ViT-L-14              | SAM2-L           | 35.7 | 33.3 | 36.8 | 34.2 |
> | DINOv3-ViT-B-16              | SAM2-L           | 32.8 | 30.9 | 33.5 | 31.6 |
> | DINOv3-ViT-L-16              | SAM2-L           | 33.8 | 32.2 | 34.4 | 32.8 |
> | DINOv3-ViT-H-16+             | SAM2-L           | 25.5 | 24.4 | 26.2 | 25.0 |
> | DINOv3-ViT-H-16+ (px)        | SAM2-L           | 26.8 | 25.5 | 27.6 | 26.3 |
> | CLIP-ViT-B-32                | SAM2-L           | 15.2 | 13.9 | 15.8 | 14.6 |
> | CLIP-ViT-B-16                | SAM2-L           | 19.2 | 18.1 | 19.5 | 18.3 |
> | CLIP-ViT-L-14                | SAM2-L           | 18.6 | 17.3 | 18.7 | 17.4 |
> | CLIP-ViT-L-14-336px          | SAM2-L           | 17.8 | 16.7 | 17.7 | 16.7 |
> | PE-Spatial-L-14-448 (PE)     | SAM2-L           | 25.7 | 24.1 | 26.5 | 24.7 |
> | PE-Spatial-L-14-448 (IN)     | SAM2-L           | 26.5 | 24.7 | 27.3 | 25.4 |
> | PE-Spatial-G-14-448 (PE)     | SAM2-L           | 24.7 | 23.1 | 24.9 | 23.3 |
>
> Overall, our pipeline transfers across backbones with moderate performance variation. CLIP models exhibit a significant performance drop, primarily due to their low-resolution and noisier feature maps. In contrast, PE-Spatial and DINOv3 achieve competitive results with only modest degradation. Since all models were used off-the-shelf without hyperparameter adjustments, we expect these backbones to benefit from more careful integration.
>
> Figure 4 visualises backbones feature maps with PCA. CLIP features show low spatial detail and irregular patterns, while PE-Spatial features are noisy but still discriminative. DINOv2 and DINOv3 produce consistent, high-resolution features, aligning with their stronger performance. Figure 5 illustrates the resulting predictions across backbones, alongside memory-bank exemplars with their corresponding PCA feature visualisations. More examples for feature and model outputs comparison are provided in Figure 15 and Figure 16 respectively.
>
> These results demonstrate that our method is not tied to a specific backbone and can operate with a range of foundation encoders, and benefit from future foundation models releases.
>
> See Appendix A.1, Table 8 and Figures 4, 5, 15, 16 for full quantitative and qualitative results.
>
> ---
> ## Comparison with VLM+SAM pipelines (Appendix A.2)
>
> We agree that including comparisons with vision–language pipelines is important, and will strengthen the manuscript. We add a direct comparison with Qwen2.5-VL-7B [1] and SAM2 pipeline, which predicts category-conditioned bounding boxes that used to prompt SAM2 to obtain segmentation masks. We evaluate both detection (nAP) and segmentation performance using the same COCO few-shot class split as our method. Table C (extended results in Appendix A.2 Table 9) shows the results obtained, highlighting the performance of our model.
>
> *Table C: Results comparing VLM+SAM2 and our method on COCO-FSOD.*
>
> | **Backbone models**                   | **Num shots** | **bbox nAP** | **segm nAP** |
> |---------------------------------------|---------------|--------------|--------------|
> | dinov2-vitl14-pretrain + sam2-hiera-l | 30            | 36.8         | 34.2         |
> | Qwen2.5-VL-7B-Instruct + sam2-hiera-l | -             | 6.2          | 5.9          |
>
> We observe that leveraging dense semantic correspondences from a vision foundation model (DINOv2) is more reliable than relying on VLM-generated boxes for instance-level segmentation in the few-shot regime. See Appendix A.2, Table 9 for full numbers.
>
> [1] Qwen2-vl: Enhancing vision-language model’s perception of the world at any resolution, Wang et al., arXiv:2409.12191

---

> ### Author Response · Authors · 2025-11-24
> **Author response [3/3]**
>
> ## Similar-class confusion (Appendix A.5):
>
> We agree with the reviewer that failure cases will benefit from a deeper semantic explanation.
>
> We compute t-SNE embeddings of object-level DINOv2 features over COCO, shown in Appendix A.5 Figure 13.
>
> We observe that easily distinguishable classes (e.g., cat–dog, airplane–bear) form well-separated clusters. In contrast, semantically similar classes (e.g., car–truck, chair–couch, wine glass–cup–vase) show substantial embedding overlap. This indicates that similar-class confusion is largely driven by DINOv2’s feature geometry rather than prototype construction.
>
> Improving performance in such cases requires stronger semantic disentanglement at the backbone level, which remains an open research direction.
>
> ---
> ## Feature similarity (Appendix A.6)
>
> To better illustrate how our feature-matching pipeline operates, we visually analyse DINOv2 feature similarity at the patch level and at the prototype level. Given a reference mask, we extract its DINOv2 features and visualize them via PCA. (see Figure 14). We then compare two forms of similarity:
> - Single-feature similarity (patch feature). We select one feature vector inside the reference mask and compute intra-class cosine similarity within the same image, and inter-class similarity with a target image. These maps often highlight only the corresponding part of the object (e.g., the neck of a dog), showing that single features capture local appearance but lack full object coverage.
> - Prototype similarity (aggregated features). We average all features inside the reference mask to obtain a class-level prototype. When computing intra-class and inter-class similarity with this prototype, similarity maps become more spatially coherent and object-aligned. This confirms that averaging features over the entire object yields a more stable representation for matching instances in the memory bank.
>
> Figure 14 shows one example comparing both the single-feature and aggregated-feature similarity. Figures 17 and 18 provide additional paired examples across more images and classes.
>
> ---
> ## Earth Mover’s Distance (Section 4.4)
> We thank the reviewer for raising this point. We have updated Section 4.4 in the manuscript to clarify this aspect and improve reader understanding.
>
> ---
> ## Reference-image degradation (Appendix A.3.3)
>
> We thank the reviewer for this suggestion. We evaluate our method under progressively degraded reference images by applying increasing levels of Gaussian blur. For efficiency, we conducted this experiment on a small representative subset, since our goal is to measure relative performance degradation rather than absolute metrics.
>
> As shown in Figure 12, our method remains robust even under strong blur. We attribute this stability to the invariance and consistency of DINOv2 features, which remain discriminative despite significant image degradation.
>
> ---
> ## Model size and GPU memory (Section 4.4)
>
> We thank the reviewer for the suggestion. We have added the requested model-size and memory-usage details to Section 4.4. Our full pipeline occupies 1.97 GiB at runtime. When using 20 classes and 10 shots per class (200 reference images) CUDA-allocated memory reaches a peak of 11.5 GiB. These additions clarify the deployment footprint and will support future extensions toward lightweight edge-oriented variants.
>
> ---
> We hope we have addressed all questions and, in light of our clarifications, we hope the reviewer *hGWi* would consider raising their score.

---

### Official Review · Reviewer_Mv7d · 2025-10-31

**Soundness:** 3
**Presentation:** 2
**Contribution:** 2
**Rating:** 6
**Confidence:** 4

**Summary:**

Existing methods—such as frameworks like SAM—generate masks that lack semantic awareness. They require manual intervention or complex prompt-generation pipelines, which creates limitations in automated processing and cross-domain scenarios. Analyses show that these existing methods need fine-tuning for novel categories, which gives rise to issues including task-specific data requirements, overfitting, and domain shift. Additionally, methods that integrate pre-trained models suffer from drawbacks in computational efficiency and instance-level segmentation capabilities.
To address these challenges, this paper proposes constructing a category-specific feature memory bank, refining feature representations through two-step aggregation, and performing feature matching with a semantic-aware soft merging strategy.

**Strengths:**

1、This method not only addresses the automation challenges of frameworks like SAM—such as their "lack of semantic awareness and need for manual intervention"—but also avoids issues like overfitting and domain shift caused by the "requirement for fine-tuning on novel categories" in traditional methods. Its effectiveness has been verified through laboratory experiments.

2、The paper achieves better optimization for scenarios (e.g., camouflaged objects) that existing vision foundation models—such as SAM (designed for segmenting everything) and DINO-V2 or CLIP (used for vision-language alignment), as illustrated in Figure 1—struggle to handle. This is beneficial for the expansion of existing methods in vertical domains.

**Weaknesses:**

1、A key limitation of traditional DINO/CLIP-based frameworks for training-free open-vocabulary semantic segmentation (OVSeg) lies in their requirement for a predefined category list during evaluation—this prevents them from being classified as genuine open-vocabulary methods. By comparison, generative vision-language model (VLM)-based methods possess intrinsic properties that make them more adept at realizing open-domain perception. Please analyze and compare the strengths of the aforementioned method (the one in question) and VLM-based approaches.

2、In the ablation study, the "Variance in Reference Set" experiment demonstrates that different reference images lead to performance variations, and increasing the number of shots (shot count) can reduce such deviations. I believe this is because the model requires certain typical, high-value references to enhance its generalization ability, while a larger shot count is more likely to improve the diversity of reference images. Is there any attempt to achieve the highest possible performance using the fewest possible high-value reference images? Alternatively, could a guiding method be proposed for selecting reference images?

3、Has there been any attempt to replace the segmenter (SAM-based) and the model used for classification (DINOv2)? This would demonstrate that the method can be plug-and-play replaced and generalized when better foundation models emerge in the future.

**Questions:**

See weakness part.

---

> ### Author Response · Authors · 2025-11-24
> **Author response [1/2]**
>
> We thank the reviewer for their valuable feedback, and respond to each point below.
>
> ---
> ## Comparison to VLM-based approaches (Appendix A.2):
>
> To compare with recent VLM models, we implement a joint pipeline that combines Qwen2.5-VL-7B [1] and SAM2. We run Qwen2.5-VL-7B on COCO validation set to produce category-conditioned bounding boxes, which are then directly used to prompt SAM2 to obtain segmentation masks. We evaluate both detection (nAP) and segmentation performance using the same COCO few-shot class split as our method. Table A shows the results obtained, highlighting the performance of our model.
>
> _Table A: Results comparing VLM+SAM2 and our method on COCO-FSOD_
>
> | **Backbone models**                   | **Num shots** | **bbox nAP** | **segm nAP** |
> |---------------------------------------|---------------|--------------|--------------|
> | dinov2-vitl14-pretrain + sam2-hiera-l | 30            | 36.8         | 34.2         |
> | Qwen2.5-VL-7B-Instruct + sam2-hiera-l | -             | 6.2          | 5.9          |
>
> We observe that leveraging dense semantic correspondences from a vision foundation model (DINOv2) is more reliable than relying on VLM-generated boxes for instance-level segmentation in the few-shot regime. See updated manuscript, Appendix A.2, Table 9 for full numbers.
>
> [1] Qwen2-vl: Enhancing vision-language model’s perception of the world at any resolution, Wang et al., arXiv:2409.12191
>
> ---
> ## Variance in reference set and selecting high-value references (Appendix A.3):
>
> We agree that performance depends on the quality of the reference images and that using “high-value” references is desirable. To address this, we conduct an analysis of reference-mask properties and derive simple, effective heuristics for selecting valuable references (Appendix A.3).
>
> *Annotation analysis.*
>
> - Mask area: distribution is heavily skewed to small objects (Figure 6).
> - Mask center location: 2D heatmaps show a strong center bias for most classes (Figure 8).
> - Distance to image edges: many masks are close to or touching image boundaries (Figure 7).
>
> *Per-class reference sensitivity.*
>
> For each novel class, we (1) sample 100 diverse reference images spanning a wide range of mask sizes, locations and edge distances, (2) evaluate each reference on a reduced validation subset. Figure 9 shows that larger mask area correlates strongly and consistently with higher mAP across all novel classes.
>
> *Heuristics for selecting high-value references.*
>
> Based on these observations, we propose automatic reference-selection rules.
>
> - Area: categorise masks by class-specific quartiles (small ≤25th, medium 25–50th, large ≥75th percentile), and prefer medium/large.
> - Centeredness: prefer masks whose centers lie within ±10% of the image center.
> - Edge avoidance: require a minimum distance ddd from all image borders.
>
> Figure 10 and Figure 11 show that medium and large masks significantly outperform small masks, and centered references are preferred over off-center ones. Furthermore, we find that once mask area and “centeredness” are controlled, the distance to frame has a weaker effect. These selection heuristics effectively improve one-shot performance without the need for increasing the number of shots, thus providing a practical and efficient method for choosing reference images. This evidence supports the validity of our approach in reducing performance variance while maintaining high-quality segmentation with a minimal reference set.
>
> (Appendix A.3.3 provides further robustness results under degraded/blurred references. As shown in Figure 12, we observe that DINOv2 features remain discriminative and our method is stable even under strong blur.)

---

> ### Author Response · Authors · 2025-11-24
> **Author response [2/2]**
>
> ## Replacing DINOv2 and SAM backbones (plug-and-play generalisation, Appendix A.1)
>
> To evaluate the transferability of our method across foundation models, we replace both the semantic encoder (DINOv2) and the SAM-based segmenter with several alternatives. We test CLIP, DINOv3, and PE-Spatial models as semantic backbones, and different SAM variants as segmentation backbones. Results are reported in Table B (full results provided in Appendix A.1, Table 8)
>
> *Table B. Backbone ablation on COCO-FSOD, comparing different semantic and segmentation backbones*
> | **Semantic backbone**        | **Segm. backb.** | **10-shot bbox nAP** | **10-shot segm nAP** | **30-shot bbox nAP** | **30-shot segm nAP** |
> |------------------------------|------------------|-----------------------|------------------------|-----------------------|------------------------|
> | DINOv2-ViT-L-14              | SAM2-L           | 35.7 | 33.3 | 36.8 | 34.2 |
> | DINOv3-ViT-B-16              | SAM2-L           | 32.8 | 30.9 | 33.5 | 31.6 |
> | DINOv3-ViT-L-16              | SAM2-L           | 33.8 | 32.2 | 34.4 | 32.8 |
> | DINOv3-ViT-H-16+             | SAM2-L           | 25.5 | 24.4 | 26.2 | 25.0 |
> | DINOv3-ViT-H-16+ (px)        | SAM2-L           | 26.8 | 25.5 | 27.6 | 26.3 |
> | CLIP-ViT-B-32                | SAM2-L           | 15.2 | 13.9 | 15.8 | 14.6 |
> | CLIP-ViT-B-16                | SAM2-L           | 19.2 | 18.1 | 19.5 | 18.3 |
> | CLIP-ViT-L-14                | SAM2-L           | 18.6 | 17.3 | 18.7 | 17.4 |
> | CLIP-ViT-L-14-336px          | SAM2-L           | 17.8 | 16.7 | 17.7 | 16.7 |
> | PE-Spatial-L-14-448 (PE)     | SAM2-L           | 25.7 | 24.1 | 26.5 | 24.7 |
> | PE-Spatial-L-14-448 (IN)     | SAM2-L           | 26.5 | 24.7 | 27.3 | 25.4 |
> | PE-Spatial-G-14-448 (PE)     | SAM2-L           | 24.7 | 23.1 | 24.9 | 23.3 |
> | DINOv2-ViT-L-14              | SAM2-T           | 27.0 | 26.1 | 27.9 | 26.9 |
> | DINOv2-ViT-L-14              | SAM2-S           | 29.8 | 27.7 | 30.6 | 28.5 |
> | DINOv2-ViT-L-14              | SAM2-B+          | 29.6 | 28.4 | 30.4 | 29.2 |
>
> Overall, our pipeline transfers across backbones with moderate performance variation. CLIP models exhibit a significant performance drop, primarily due to their low-resolution and noisier feature maps. In contrast, PE-Spatial and DINOv3 achieve competitive results with only modest degradation. Since all models were used off-the-shelf without hyperparameter adjustments, we expect these backbones to benefit from more careful integration.
>
> Figure 4 (Appendix A.1) visualises backbones feature maps with PCA. CLIP features show low spatial detail and irregular patterns, while PE-Spatial features are noisy but still discriminative. DINOv2 and DINOv3 produce consistent, high-resolution features, aligning with their stronger performance. Figure 5 illustrates the resulting predictions across backbones, alongside memory-bank exemplars with their corresponding PCA feature visualisations. More examples for feature and model outputs comparison are provided in Figure 15 and Figure 16 respectively.
>
> These results demonstrate that our method is not tied to specific backbone and can operate with a range of foundation encoders, and benefit from future foundation models releases.
>
> See Appendix A.1, Table 8 and Figures 4, 5, 15, 16 for full quantitative and qualitative results.
>
> ---
> We hope these clarifications and new experiments address the reviewer’s concerns.

---

### Author Response · Authors · 2025-11-24

We thank all reviewers for their thoughtful feedback.

We have updated the manuscript accordingly, with all changes highlighted in blue.

Please find our replies to the reviewers below.

---

### Author Response · Authors · 2025-12-01
**Summary Post to AC/SAC**

We sincerely thank the AC/SAC for their efforts.

Below, we provide a concise summary of our rebuttal and the discussion before the freeze. For full details, please refer to our individual responses to each reviewer.

---

## Summary of the rebuttal

We significantly expanded our experimental validation and analysis (Appendix A.1 - A.8) in light of reviewer comments. Key additions include:

- **Reviewer Mv7d**:
  - Added systematic comparison to VLM+SAM pipelines (Qwen2.5-VL-7B + SAM2), showing clear advantages.
  - Conducted a detailed study on reference-image quality, establishing actionable heuristics (mask area, centeredness, frame proximity).
  - Added extensive experiments on backbone-replacement experiments, demonstrating plug-and-play generalisation.

- **Reviewer hGWi**:
  - Added ablations for two-step vs. class-only aggregation (Table 10).
  - Added semantic-backbone ablation (CLIP, DINOv3, PE-Spatial).
  - Added failure-case analysis, including t-SNE of DINOv2 embeddings and patch- vs prototype-level similarity maps.
  - Added robustness tests under blurred references.
  - Added runtime, memory, and practical deployment details.

- **Reviewer 5kET**:
  - Expanded discussion of generalist models (T-REX, SINE, DINO-X).
  - Further details on pipeline deployment.
  - Added extended semantic-backbone ablation (CLIP, DINOv3, PE-Spatial).
  - Strengthened the analysis of reference ambiguity with new quantitative patterns and selection heuristics.
  - *After reviewer engagement*:
    - Added detailed feature-distribution visualisations and prediction comparisons across backbones (Figures 19–21).

---
## Reviewer engagement:

We were able to **engage with reviewer 5kET, who responded positively**:
> “I am pleased to see the broader comparison across different semantic backbones, which significantly strengthens the empirical evaluation of the method.”

Reviewer 5kET asked for further clarification:
- (1) “further analyse” DINOv3 vs DINOv2 differences
- (2) “visualisations of feature distributions”
- (3) “qualitative comparisons of predictions”.

We addressed all three with additional analyses:
- (1) t-SNE embeddings showing that class separability remains unchanged across DINOv2 and DINOv3. (Figures 19-20)
- (2) Feature visualisations across backbones (Figures 14, 17, 18)
- (3) Backbone output comparisons (Figures 16, 21)


The other two reviewers (Mv7d, hGWi) did not respond before the discussion was frozen. However, we provided detailed responses and substantial new experiments addressing each of their concerns. We hope that these would have been well received had more time been available.

---
## Final note

We deeply appreciate the AC/SAC’s work under these circumstances. Thank you for your time and consideration!

---

### Meta-Review · Area_Chair_kxrd · 2026-01-07

**Summary:**

This submission was reviewed by three expert reviewers, with the ratings of: 2 borderline accept, and 1 borderline reject. The major concerns from the reviewers are around differences from VLM-based approaches, selection of reference images, generalization, technical novelty, insufficient experimental analysis, limited comparison to closely related works, and evaluation scope. Rebuttal was provided by the authors to address the concerns, and there was one reviewer engaged in the follow-up discussion.

After carefully going through the review comments, the authors' rebuttal and the discussions, it can be seen that some of the concerns and questions are addressed by the authors' further response and experiments. However, there are still major concerns remaining not well addressed. There is no strong support for a clear accept. As a result, it is unfortunate that the paper in its current form is not ready to be presented at ICLR, and needs more than a major revision followed by another round of review for assessment. However, considering the interesting findings in this paper, the authors are encouraged to further revise their paper according to the review comments and consider submitting to a future venue.

**Reviewer Concerns:**

Concerns that the AC thinks were addressed by the rebuttal: comparison with VLM-based approaches; replacement of the segmented and classification model; comparison to VML+SAM; failure cases and visualisation issues; different encoders comaprison.

Concerns that are still outstanding: the selection of reference images; the limited technical novelty and contributions, though with the further explanation; the effectiveness validation from the two-step aggregation study and the memory bank design; evaluation scope; and the backbone size phenomena.

**Reviewer Scores:**

According to the review comments, and the rebuttal, for each review the reviewer might have changed their score in the way below, if they had been able to participate fully in the discussion:
* Reviewer Mv7d: borderline accept to borderline reject, or unchanged
* Reviewer hGWi: borderline reject to reject, or unchanged
* Reviewer 5kET: borderline accept to borderline reject, or unchanged.

---

### Decision · Program_Chairs · 2026-01-26

Reject